# Beyond Markov Assumption: Improving Sample Efficiency in MDPs by Historical Augmentation

## Abstract

Under the Markov assumption of Markov Decision Processes (MDPs), an optimal stationary policy does not need to consider history and is no worse than any non-stationary or history-dependent policy. Therefore, existing Deep Reinforcement Learning (DRL) algorithms usually model sequential decision-making as an MDP and then try to optimize a stationary policy by single-step state transitions. However, such optimization is often faced with sample inefficiency when the causal relationships of state transitions are complex. To address the above problem, this paper investigates if augmenting the states with their historical information can simplify the complex causal relationships in MDPs and thus improve the sample efficiency for DRL. First, we demonstrate that a complex causal relationship of single-step state transitions may be inferred by a simple causal function of the historically augmented states. Then, we propose a convolutional neural network architecture to learn the representation of the current state and its historical trajectory. The main idea of this representation learning is to compress the high-dimensional historical trajectories into a low-dimensional space. In this way, we can extract the simple causal relationships from historical information and avoid the overfitting caused by high-dimensional data. Finally, we formulate Historical Augmentation Aided Actor-Critic (HA3C) algorithm by adding the learned representations to the actor-critic method. The experiment on standard MDP tasks demonstrates that HA3C outperforms current state-of-the-art methods in terms of both sample efficiency and performance.

## 1 Introduction

Sequential decision-making widely exists in real-world control tasks, such as robot control and autonomous driving (Dorf & Bishop, 2011; Ibarz et al., 2021; Sallab et al., 2017). Generally speaking, it can be modelled as a Markov Decision Process (MDP), where an agent iteratively takes action in an environment for transiting from one state to another (Puterman, 1990). Each transition is evaluated by a reward signal passing from the environment to the agent so that Reinforcement Learning (RL) can learn the optimal policy by maximizing the cumulative reward (Sutton & Barto, 2018). The Markov Assumption of MDPs asserts that the probability distributions of the reward and next state depend only on the current state and action. Under the Markov assumption of MDPs, there exists an optimal stationary policy which does not need to consider history and is no worse than any non-stationary or history-dependent policy (Puterman, 2014). Therefore, existing RL algorithms usually try to optimize a stationary policy by single-step state transitions.

With advances in deep learning, many effective Deep RL (DRL) methods were proposed (Fujimoto et al., 2018; Haarnoja et al., 2018; Lillicrap et al., 2016; Mnih et al., 2016; 2015). Under the Markov assumption of MDPs, they are usually based on the actor-critic method where the critic estimates the $Q$-function, i.e., the expected cumulative reward after taking action at each state, while the actor updates the policy to choose the action which can maximize the estimated $Q$-function (Schulman et al., 2015; Silver et al., 2014). However, such optimization may miss the useful causal relationships of state transitions, leading to sample inefficiency (Allen et al., 2021; Buckman et al., 2018; Du et al., 2020; Guo et al., 2020). An existing partial solution to this issue is representation learning in which a neural network is trained to infer the causal relationships of state transitions by predicting the

reward or future state of each state-action pair (Munk et al., 2016; Ni et al., 2023; Ravindran, 2004; Rezaei-Shoshtari et al., 2022). Then, the sample efficiency of DRL can be improved by adding the learned representations to the actor-critic method. Unfortunately, it is hard to train the neural networks which can infer complex causal relationships, e.g., polynomial causal relationships and the basic laws of physics (Andoni et al., 2014; Cranmer et al., 2020). Standard complexity-theoretic results strongly suggest that there is no algorithm efficient enough for learning arbitrary target functions, even for target functions representable by very low-depth networks (Applebaum et al., 2006). Therefore, the sample efficiency for DRL is still limited in complex MDP tasks.

This paper addresses the above problem by augmenting the states with their historical information. Based on the analysis in Section 3, we believe that historical augmentation can simplify the causal relationships of state transitions by its inherent contextual information and increasing the search space of the causal functions (Hallak et al., 2015; Sprunger & Jacobs, 2019). Therefore, we focus on optimzing a history-dependent stationary policy in an MDP. Our DRL approach comprises two key components: 1) Learning the state representations to capture the causal relationships in an MDP and 2) finding the optimal stationary policy by the learned representations. Given an action and the historically augmented current state, our representation learning utilizes a Convolutional Neural Network (CNN) architecture to compress the high-dimensional historical trajectory of the given state into a low-dimensional space while predicting the future state. The compressed historical trajectories can be seen as the abstracted features which can represent the simple causal relationships and avoid the overfitting caused by high-dimensional data (Andre & Russell, 2002). To keep the Markov assumption of MDPs, our representation learning does not compress the current state. We add the learned state representations to the actor-critic method. In this way, the causal relationships captured by our representation learning can be utilized to estimate the $Q$-function and update policy. Therefore, our new DRL approach can optimize the policy in a complex MDP with high sample efficiency. We combine historical augmentation, state representations, and TD3 in our approach to formulate a new DRL algorithm, Historical Augmentation Aided Actor-Critic (HA3C). The experiment on standard MDP tasks, i.e. Mujoco control tasks and Deep Mind Control (DMC) suite, empirically demonstrates that HA3C outperforms current state-of-the-art methods in terms of both sample efficiency and performance (Brockman et al., 2016; Todorov et al., 2012; Tassa et al., 2018).

Our contributions are as follows: 1) Existing RL methods usually utilize historical information to recover Markov assumption in dynamics. It is the first time in the literature that historical augmentation can be used to improve sample efficiency when Markov assumption is satisfied. 2) We propose a new DRL approach to address the problem of how to effectively utilize the historical information in MDPs. 3) Based on this approach, we formulate a new RL algorithm, HA3C, which outperforms existing state-of-the-art DRL algorithms, e.g. TD7 (Fujimoto et al., 2023). 4) Our examples, experiment, and discussion illustrate that in fact, DRL needs to consider historical information in complex MDP tasks.

## 2 BACKGROUND

An MDP can be written as a 5-tuple $\mathbb{M} = \langle \mathcal{S}, \mathcal{A}, R, \boldsymbol{P}, \gamma \rangle$ with state space $\mathcal{S}$, action space $\mathcal{A}$, reward function $R$, transition model $\boldsymbol{P}$, and discount factor $\gamma$. In an MDP, RL can maximize the discounted cumulative reward by learning how to map the states to the actions (Baird, 1995; Duan et al., 2016; Williams, 1992). For a given state $\boldsymbol{s}_t \in \mathcal{S}$ at time step $t$, the agent executes an action $\boldsymbol{a}_t \in \mathcal{A}$ w.r.t. a policy $\pi : \mathcal{S} \mapsto \mathcal{A}$, to obtain a reward $r_t = R(\boldsymbol{s}_t, \boldsymbol{a}_t)$ and transfer to a new state $\boldsymbol{s}_{t+1}$. The return of the agent is defined as the discounted cumulative reward $G_t = \sum_{i=t}^{+\infty} \gamma^{i-t} r_i$. Based on the Markov assumption of MDPs, RL can find the optimal policy to maximize the following value function which is the expected return when $\boldsymbol{s}_t = \boldsymbol{s}$ and following $\pi$ thereafter.

$$V^\pi(\boldsymbol{s}) = \mathbb{E}^\pi \left[ G_t | \boldsymbol{s}_t = \boldsymbol{s} \right] = \mathbb{E}^\pi \left[ \sum_{i=0}^{+\infty} \gamma^i r_{t+i} | \boldsymbol{s}_t = \boldsymbol{s} \right],$$

where $\mathbb{E}^\pi[*]$ denotes the expected value of a random variable given that the agent follows policy $\pi$.

With advances in deep learning, combining neural networks into RL has drawn significant attention in the literature. Many DRL algorithms learn the optimal policy by the actor-critic method (Kaelbling et al., 1996), where the critic network estimates the $Q$-function which is the expected return when

$s_t = s$, $a_t = a$, and following policy $\pi$ thereafter.

$$Q^\pi(s, a) = \mathbb{E}^\pi \left[ G_t | s_t = s, a_t = a \right] = \mathbb{E}^\pi \left[ \sum_{i=0}^{+\infty} \gamma^i r_{t+i} | s_t = s, a_t = a \right],$$

while the actor network updates the policy to maximize the estimated $Q$-function.

To improve sample efficiency, some DRL methods learn the state representations of the collected state transitions and then add the learned representations to the actor-critic method (Anand et al., 2019; Dayan, 1993; Gelada et al., 2019; Li et al., 2006). This representation learning aims to capture the causal relationships in MDPs, and thus improves sample efficiency (Liu et al., 2020; Van Hoof et al., 2016; Ye et al., 2023; Zhang et al., 2021). For example, ML-DDPG learns the state representations by predicting the next state representation and the reward in MDPs (Munk et al., 2016). As an improvement of ML-DDPG, OFENet learns the high-dimensional state representations by predicting the next state in DenseNet architecture (Ota et al., 2020). TD7 improves the learning of state representations by AvgL1Norm and then combines the learned representations with TD3, checkpoints, and prioritized replay buffer (Fujimoto et al., 2023).

DRL algorithms need to consider historical information when the Markov assumption of MDPs is violated (Eysenbach et al., 2020; Majeed & Hutter, 2018; Hafner et al., 2019b). For Partially Observable MDPs (POMDPs), in which the states are partially observable, deep recurrent $Q$-network uses LSTMs to encode state transition trajectories in the $Q$-function estimation (Hausknecht & Stone, 2015). As an improvement of deep recurrent $Q$-network, deep transformer $Q$-network uses transformers to encode the state transition trajectories (Esslinger et al., 2022). As a famous DRL algorithm, Dreamer encodes the historical information into the state at every time step in POMDPs (Ha & Schmidhuber, 2018; Hafner et al., 2019a). In delayed MDPs, in which the current state and reward may arrive at the agent with a delay (Katsikopoulos & Engelbrecht, 2003), researchers usually recover the Markov assumption of MDPs by considering the historical actions (Bouteiller et al., 2020; Derman et al., 2021). When the Markov assumption of MDPs is violated by the state abstraction, it is possible to find a history-based policy which works in the abstracted space and is of the same quality as optimal policy (Patil et al., 2024). However, the history-based DRL for the dynamics which are under Markov assumption is largely absent from the literature.

## 3 MOTIVATION

Let $h_t = \{s_0, a_0, ..., s_t\}$ as the history up to time step $t$ in a sequential decision-making task. The optimal policy may change the decision rule in different time steps and select actions based on historical information. In this case, we should optimize a history-dependent policy $\pi = \{d_t | t = 0, 1, ...\}$ which selects action at time step $t$ by decision-rule $d_t(a_t | h_t)$. Fortunately, based on the Markov assumption of MDPs, there is an optimal stationary policy $\pi(a_t | s_t)$ which is unrelated to time and selects action $a_t$ by only the state $s_t$. This Markov assumption asserts that the probability distributions of state $s_{t+1}$ and reward $r_t$ depend only on the $s_t$ and $a_t$ as

$$P\{s_{t+1} = s', r_t = r | s_0, a_0, r_0, ..., s_t, a_t = P\{s_{t+1} = s', r_t = r | s_t, a_t\},$$

where $P$ is the probability distribution in $\boldsymbol{P}$. Let $HR$ and $SR$ denote the class of history-dependent and stationary policies, respectively. Lemma 3.1 is the key of most existing RL algorithms (Puterman, 2014)[Thm. 6.2.10]. The different types of policies are detailed in Appendix A.

**Lemma 3.1.** *Under the Markov assumption of MDPs, there exists a policy $\pi^* \in SR$ such that, for all $t$, $V_{\pi^*}(s_t) = \sup_{\pi \in HR} V_\pi(h_t)$.*

Based on Lemma 3.1, existing DRL algorithms for MDPs usually optimize a stationary policy by single-step transitions. If the causal relationships in the modelled MDP are simple, e.g., there are only linear causal relationships in this MDP, such optimization effectively finds the optimal policy. A classical result is that a neural network with a single hidden layer can successfully learn a linear function (Andoni et al., 2014). However, it is still hard to capture complex causal relationships by neural networks. Standard complexity-theoretic results strongly suggest that there is no algorithm efficient enough for learning arbitrary functions, even for target functions representable by very low-depth networks (Applebaum et al., 2006). In fact, a more complex causal function requires neural networks to approximate with more parameters, samples, and time consumption (Bianchini & Scarselli, 2014).

Historical augmentation has the potential to address the above problem by simplifying the causal relationships in MDPs as it can increase the search space of the causal functions and provide much contextual information on state transitions (Hallak et al., 2015; Sodhani et al., 2022).

*Example* 3.1. For example, if we model the state transitions with Fibonacci sequence as $s_0 = 1$, $s_1 = 1$, $s_2 = 2$, $s_3 = 3$, $s_4 = 5, \cdots$, when $t > 2$, the state transitions in this model will satisfy the Markov assumption of Markov Processes as (Dynkin, 1965)

$$P\{s_{t+1} = s'|s_0, ..., s_t\} = P\{s_{t+1} = s'|s_t\}.$$

Without considering history, at $s_t$, $s_{t+1}$ will be predicted by a complex time-related formula

$$s_{t+1} = \frac{1}{\sqrt{5}}\left[\left(\frac{1+\sqrt{5}}{2}\right)^{t+1} - \left(\frac{1-\sqrt{5}}{2}\right)^{t+1}\right].$$

Fortunately, when considering history, we can predict $s_{t+1}$ by a simple linear function

$$s_{t+1} = s_{t-1} + s_t.$$

In Appendix B, we give another example to illustrate that by historical augmentation, a non-linear causal relationship in single-step transitions may be simplified as a linear causal relationship. Fig. 1(a) presents the original MDP causal relationships and Fig. 1(b) demonstrates the MDP causal relationships with state augmentation. When inferring the causal relationships in a trajectory, the causal

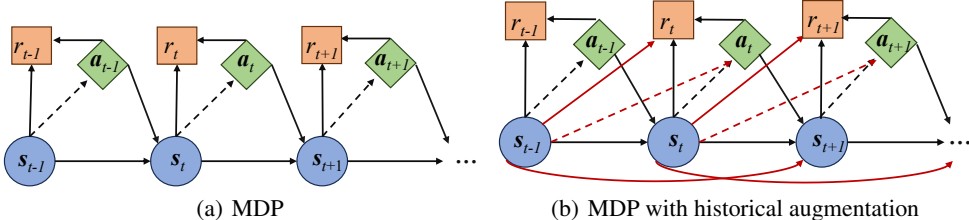

(a) MDP  (b) MDP with historical augmentation

Figure 1: Causal diagrams of an MDP with or without historical augmentation. The black lines index the original MDP causal relationships and the red lines index the added causal relationships, e.g., the causal relationships from historical augmentation.The dashed lines indicate the information needed in the optimization.

function in Fig 1(b) can be simpler than the causal function in Fig 1(a).

From the analysis above, the motivation of our work is that historical information can simplify the complex causal relationships in MDPs and thus has the potential to improve the sample efficiency of DRL. However, the challenges are 1) how to ensure that the causal relationships learned from historical augmentation are simple and 2) avoiding overfitting caused by the high-dimensional historical data.

## 4 METHOD

In this section, we propose a new DRL approach by the representation learning of historically augmented states. Then, we formulate a new DRL algorithm, HA3C, and illustrate the advantage of this algorithm with a visual example.

### 4.1 REPRESENTATION LEARNING ON HISTORICALLY AUGMENTED STATES

To address the problem of how to effectively utilize the historical information in MDPs, we propose a new DRL approach by the representation learning of historically augmented states. The main idea of this representation learning is to compress the high-dimensional historical trajectories into a low-dimensional representation space (Andre & Russell, 2002; Li et al., 2006). On the one hand, the compressed historical trajectories can be seen as the abstracted features of the historical information to extract the simple causal relationships. On the other hand, this compression will avoid the overfitting caused by the high-dimensional historical data (Ying, 2019).

To keep the Markov assumption of MDPs, our representation learning does not compress the current state. Let $s_{k,t} = \{s_{t-k+1}, ..., s_t\}$. If $t < k$, one can set each $s_i \in s_{k-t,-1}$ by the zero vector $\mathbf{0}$. The causal diagram of MDP with our state abstraction is in Fig. 2. As we can see, when predicting $s_{t+1}$ by $s_{k,t}$ and $a_t$, the dimensionality reduction is only performed on $s_{k-1,t-1}$.

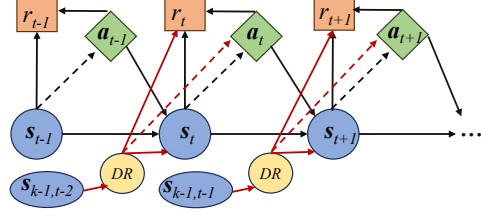

Figure 2: Causal diagram of a historically augmented MDP with state abstraction. DR represents the operation of dimensionality reduction.

Let $S_k D$ denote the class of the stationary deterministic policies based on $k$-order state trajectories. Theorem 4.1 forms the basis of our DRL approach. This theorem can be implied by Lemma 3.1. For completeness, we provide a proof in Appendix D.

**Theorem 4.1.** *Under the Markov assumption of MDPs, there exists a stationary deterministic policy $\mu^* \in S_k D$ such that, for all $t$, $V^{\mu^*}(s_{k,t}) = \sup_{\pi \in HR} V^{\pi}(h_t)$.*

To capture the simplified causal relationships in MDPs by historical augmentation, we define a pair of encoders $z^{s_{k,t}} = f(s_{k,t})$ and $z^{s_{k,t},a_t} = g(z^{s_{k,t}}, a_t)$. Based on the Markov assumption in MDPs, we can predict $z^{s_{k,t+1}}$, i.e., the representation of $s_{k,t+1}$, by $z^{s_{k,t},a_t}$. Thus, the two encoders are trained by minimizing the following predicting loss:

$$L(f,g) = ||g(f(s_{k,t}), a_t) - |f(s_{k,t+1})|_{\times}||_2^2 = ||z^{s_{k,t},a_t} - |z^{s_{k,t+1}}|_{\times}||_2^2, \tag{1}$$

where $|*|_{\times}$ denotes the stop-gradient operation. As presented in Fig. 3, a simple yet effective CNN network architecture is utilized in our representation learning. In the network of $f(s_{k,t})$, we first use a CNN layer to mine the historical information in $s_{k-1,t-1}$. This layer produces the feature maps of $s_{k-1,t-1}$ by the multiple filters, which are as wide as the state dimensionality. Second, we utilize a max pooling layer to capture the most important features and an average pooling layer to capture the tendency features. Third, we compress the captured features into a low-dimensional space and learn the features of $s_t$. Finally, we concatenate the compressed features of $s_{k-1,t-1}$ and the learned features of $s_t$. The concatenated features are the input of the next fully connected layer to obtain the representation $z^{s_{k,t}}$. In the network of $g(z^{s_{k,t}}, a_t)$, we put the concatenation of $z^{s_{k,t}}$ and $a_t$ into the two fully connected layers to obtain the representation $z^{s_{k,t},a_t}$.

We combine our learned representations with the actor-critic method and thus the $Q$-function can be defined as $\hat{Q}(z^{s_{k,t}}, a_t)$ and the policy can be defined as $\mu(z^{s_{k,t}}) \in S_k D$. Define the probability distribution of $z^{s_{k,t+1}}$ under $\mu$ as

$$P^{\mu}\{z^{s_{k,t+1}} = z^{s'_{k,:}} | z^{s_{k,t}} = z^{s_{k,:}}\} = \int_{\mathcal{Z}} \mathbb{E}_{a \sim \mu(z^{s_{k,:}})} \left[ p(z^{s'_{k,:}} | z^{s_{k,:}}, a) \right] dz^{s_{k,:}},$$

where $s_{k,:}$ is a $k$-order state trajectory $\{s_0, ..., s_{k-1}\}$ ending with $s$, i.e., $s_{k-1} = s$, $\mathcal{Z}$ is the set of all possible $z^{s_{k,:}}$, and $p(z^{s'_{k,:}} | z^{s_{k,:}}, a)$ is the probability of transferring to $z^{s'_{k,:}}$ with taking $a$ at $z^{s_{k,:}}$. Our optimization is based on a Bellman optimality operator $B$ for $\mu$ as

$$B_{\mu}\hat{Q}(z^{s_{k,:}}, a) = \max_{\mu} \mathbb{E}_{a_{t+1} \sim \mu, z^{s_{k,t+1}} \sim P^{\mu}}[r_t + \gamma \hat{Q}(z^{s_{k,t+1}}, a_{t+1})]. \tag{2}$$

The following theorem gives the conditions to find the optimal stationary policy in our approach. The proof of this theorem is given in Appendix D.

**Theorem 4.2.** *Given a finite MDP, if 1) $f(*)$ and $g(*)$ are fixed, 2) $\forall s_{k,:}, s'_{k,:} \in \mathcal{S}_{k,:}, s \neq s' \Leftrightarrow z^{s_{k,:}} \neq z^{s'_{k,:}}$, and 3) $L(f,g) \to 0$, then $\hat{Q}(z^{s_{k,t}}, a_t)$ converges to the optimal $Q^*(s_t, a_t)$ by the Bellman optimality operator in equation 2.*

This theorem illustrates that no matter whether different historical trajectories lead to different representations on the state $s$, we can still find the optimal stationary policy in the representation space. To make condition 2) hold, we can increase the dimensionality of $s$ in representation learning. This operation also can improve sample efficiency (Ota et al., 2020). To see condition 3) hold, there should exist a $s'_{k,:}$ that satisfies

$$p\{s_{k,t+1} = s'_{k,:} | s_{k,t}, a_t\} \to 1.$$

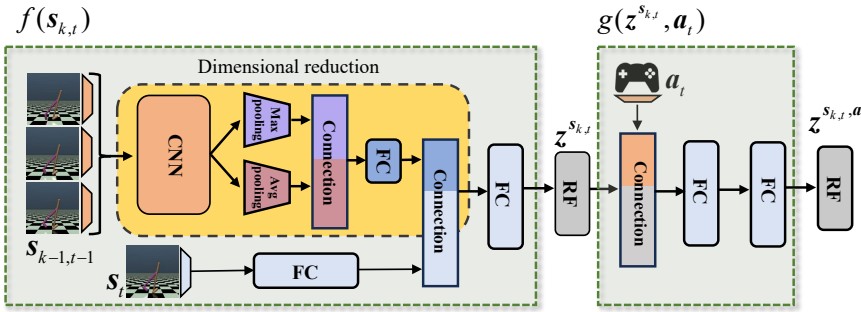

Figure 3: Network architecture of our representation learning. FC represents a fully connected layer and RF represents the state representation features.

There is an analysis of the function approximation error in Appendix D. We add $z^{s_{k,t},a_t}$ to $\hat{Q}$ to consider the learned relationship between $a_t$ and $z^{s_{k,t}}$ in the representation space. We also add $s_t$ to $\hat{Q}$ and $\mu$ to consolidate the relationships in single-step transitions. Thus $Q$ and $\mu$ can be writtern as $\hat{Q}(z^{s_{k,t},a_t}, z^{s_{k,t}}, s_t, a_t)$ and $\mu(z^{s_{k,t}}, s_t)$, respectively. The operations in $\hat{Q}$ and $\mu$ are shown in Fig. 4.

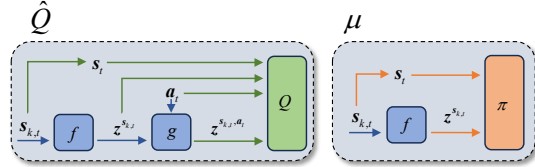

Figure 4: The operations in $Q$ and $\mu$.

Our approach can be connected with POMDPs, High-order MDPs (HMDPs), and state abstraction. A detailed analysis of the connections between our approach and the related work is shown in Appendix C.

## 4.2 HA3C ALGORITHM

In this subsection, we propose HA3C algorithm which is a combination of TD3, representation learning, historical augmentation. HA3C has several networks as follows. Two critic networks $(\hat{Q}_{\phi_1}, \hat{Q}_{\phi_2})$, two target critic networks $(\hat{Q}_{\phi_1^T}, \hat{Q}_{\phi_2^T})$, an actor network $\mu_\theta$, a target actor network $\mu_{\theta^T}$, two encoders $(f_\sigma, g_\sigma)$, two fixed encoders $(f_{\sigma^F}, g_{\sigma^F})$, two target encoders $(f_{\sigma^T}, g_{\sigma^T})$, a checkpoint actor network $\pi_{\theta^C}$, and a checkpoint encoder $f_{\sigma^C}$.

To learn the representations with historical augmentation, $f_\sigma$, and $g_\sigma$ are trained by the transitions in buffer $\mathcal{B} = \{s_{k,i}, a_i, r_i, s_{k,i+1}\}$ to minimize the predicting loss in equation 1. For any parameter set $\alpha$, we define

$$z_\alpha^{s_{k,t}} = f_\alpha(s_{k,t}), \quad z_\alpha^{s_{k,t},a_t} = g_\alpha(z^{s_{k,t}}, a_t).$$

Based on the assumption that $f_{\sigma^F}$ and $g_{\sigma^F}$ satisfy the conditions in Theorem 4.2 on the most transitions in $\mathcal{B}$, the $Q$-function is estimated by the following Huber loss function (Huber, 1992).

$$L(\phi_i, \mathcal{B}) = Huber_{(s_{k,t},a_t,r_t,s_{k,t+1})\sim\mathcal{B}}\left[x_t - (\hat{Q}_{\phi_i}(z_{\sigma^F}^{s_{k,t},a_t}, z_{\sigma^F}^{s_t}, s_t, a_t)\right], \quad (3)$$

$$x_t = r_t + \gamma clip(\min(\hat{Q}_{\phi_i^T}(z_{\sigma^T}^{s_{k,t+1},a'}, z_{\sigma^T}^{s_{t+1}}, s_{t+1}, a')), \hat{Q}^{\min}, \hat{Q}^{\max}),$$

$$a' = \mu_{\theta^T}(z_{\sigma^T}^{s_{k,t+1}}, s_{t+1}) + \epsilon_T, \epsilon_T \sim \mathcal{N},$$

where $\epsilon_T$ is target policy noise (Fujimoto et al., 2018), $\mathcal{N}$ is a Gaussian distribution $\mathcal{N}(0, \sigma)$, and $\hat{Q}^{\min}$ and $\hat{Q}^{\max}$ are updated at each time step as

$$\hat{Q}^{\max} \leftarrow \max(x_t, \hat{Q}^{\max}), \quad \hat{Q}^{\min} \leftarrow \min(x_t, \hat{Q}^{\min}).$$

Based on the learned $Q$-function, the policy network $\pi_\theta$ is updated by

$$\max_\theta \mathbb{E}_{s_{k,t}\sim\mathcal{B}}\left[\sum_{i=1,2} \hat{Q}_{\phi_i}(z^{s_{k,t},a}, z^{s_t}, s_t, a)\right], \quad (4)$$

$$a = \mu_\theta(z_{\sigma^F}^{s_{k,t}}, s_t).$$

To explore the new actions and thus generate new transitions in $\mathcal{B}$, exploration noise $\epsilon$ is added as

$$\boldsymbol{a}_t \leftarrow \boldsymbol{a}_t + \epsilon_e, \epsilon_e \sim \mathcal{N}.$$

In our TD learning, $\sigma^F$, $\sigma^T$, $\phi^T$, and $\theta^T$ are updated by

$$\sigma^F \leftarrow \sigma^T, \quad \sigma^T \leftarrow \sigma, \quad \phi^T \leftarrow \phi, \quad \theta^T \leftarrow \theta. \tag{5}$$

Because DRL algorithms are unstable (Henderson et al., 2018; Teh et al., 2017), we use the checkpoint policy to obtain the cumulative reward in our evaluation (Vaswani et al., 2017). In the training of HA3C, if the current policy outperforms the checkpoint policy, we will update the checkpoint policy with the current policy, then $\sigma^C \leftarrow \sigma$ and $\theta^C \leftarrow \theta$. The checkpoint policy can give a more accurate evaluation by maintaining the high-performance policy unchanged. Furthermore, the LAP replay buffer is utilized to store and replay the transitions (Fujimoto et al., 2023; 2020). The algorithm of online HA3C is presented in Algorithm 1.

---

**Algorithm 1** Online HA3C

---

Initialize the hyper-parameters and networks
Initialize replay buffer $\mathcal{B}$
**for** $episode = 0$ to $episode_{max}$ **do**
    Collect $k$-order transitions by $\mu_\theta$ and store them in LAP buffer $\mathcal{B}$
    **if** Checkpoint condition **then**
        **if** $\mu_\theta$ outperforms $\mu_{\theta^c}$ then **then**
            Update checkpoint networks $\mu_{\theta^c} \leftarrow \mu_\theta$ and $f_{\sigma^c} \leftarrow f_\sigma$
        **end if**
    **end if**
    Sample $k$-order transitions from LAP buffer $\mathcal{B}$
    Train the encoder $f_\sigma$ and $g_\sigma$ by equation 1
    Train $\hat{Q}_{\phi_1}$ and $\hat{Q}_{\phi_2}$ by equation 3
    Train $\pi_\theta$ by equation 4
    **if** Target update frequency steps have passed **then**
        Update target networks by equation 5
    **end if**
**end for**

---

Fig. 5 is an example to illustrate the advantage of learning the policy in HA3C. We first collect the obtained states of Walker2d MuJoCo control task by learning the policy with and without historical augmentation, respectively. The max learning step is $4 \times 10^5$. Then we map the collected states in 2D space together by UMAP. Finally, we show the reached states without the learning of historical augmentation in the left subfigure of Fig. 5 and the reached states with the learning of historical augmentation in the right subfigure of Fig. 5. Each state is coloured by the reward of reaching it. As we can see, although the actions to obtain the states in high-reward regions (indexed by the red circles) can be explored, without historical augmentation, it is hard to learn the policy which can regenerate these explored actions. Therefore, in the left

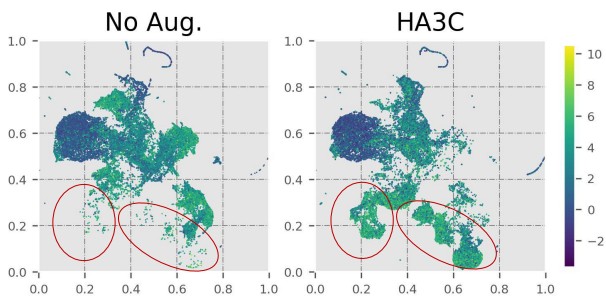

Figure 5: Visual results of the obtained states in Walker2d environment. Each state is coloured by the reward of reaching this state.

subfigure, there are only a few states in the high-reward regions. Fortunately, as shown in the right subfigure, there are a lot of states in the high-reward regions when learning the policy with historical augmentation. The visual results of other environments are shown in Appendix F.

# 5 EXPERIMENTAL RESULT

In this section, first, we compare HA3C to five existing RL algorithms on five Mujoco control tasks (Todorov et al., 2012). Second, we give the ablation study of HA3C to illustrate that historical augmentation is the real source of the improvement in sample efficiency. Third, we analyze the parameter sensitivity on the length of the historical state trajectory and the number of dimensions of compressed historical trajectories. Finally, we give the running times of the different RL algorithms. The experimental setting is in Appendix E. Appendix F has some supplementary experiments including the state visualization and DMC experiment (Tassa et al., 2018).

## 5.1 COMPARATIVE EVALUATION

In this subsection, we evaluate our HA3C on five MuJoCo control tasks including Walker2d, HalfCheetah, Ant, Humanoid, and Hopper. The compared algorithms are TD3 (Fujimoto et al., 2018), SAC (Haarnoja et al., 2018), TQC (Kuznetsov et al., 2020), TD3+OFE (Ota et al., 2020), and TD7 (Fujimoto et al., 2023). For all algorithms, each task runs 10 instances with different random seeds. In each instance, the evaluation is performed every 5000 time steps. The learning curves are shown in Fig. 6 and the numerical results at 400K time step and 1M time step are shown in Table 1.

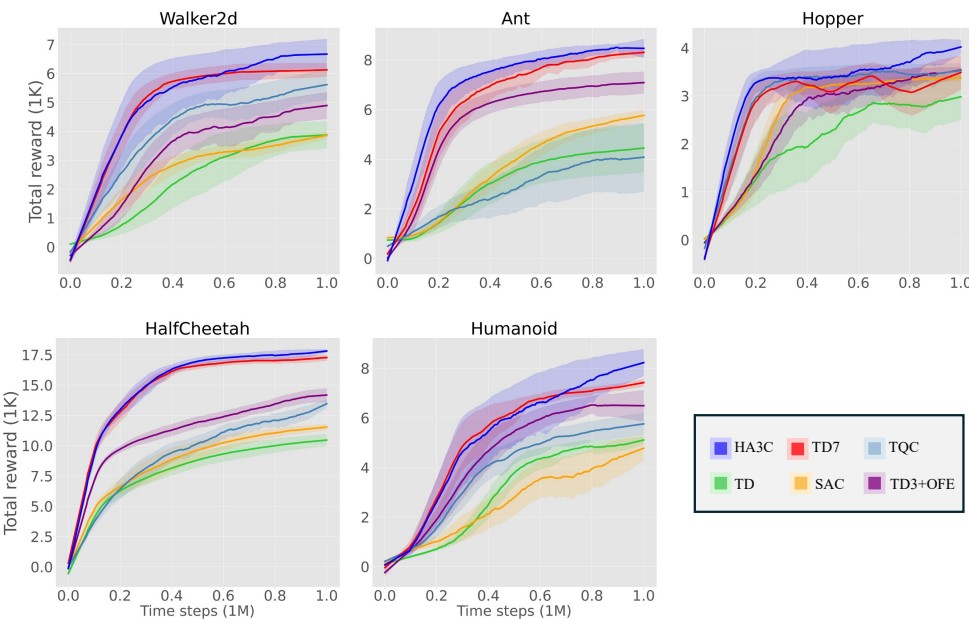

Figure 6: Learning curves of different RL algorithms on the MuJoCo control tasks. The shaded area captures a 90% confidence interval around the average performance.

From Fig. 6 and Table 1, we can see that 1) With the help of historical augmentation, HA3C significantly outperforms the compared algorithms in terms of the early average highest returns (400K time step) and final average highest returns (1M time step); 2) as shown in Fig. 6, because of the instability in rapidly learning complex causal relationships, the early average returns of HA3C on Walker2d and Humanoid are a little lower than the early average returns of TD7, however, HA3C can get the highest final average returns on all of the control tasks.

## 5.2 ABLATION STUDY

Our ablation study aims to prove that our historical augmentation is the real source of the improvement in sample efficiency. Therefore, we compare HA3C to the following two ablations: 1) Copy Aug. copies the current state $k$ times instead of augmenting with $k$ steps of history in our CNN; 2) No Aug. is TD3 with single-step representation learning and LAP. Our ablation study is performed on Ant, Hopper, and Walker2d. All of the comparison methods have the same parameter setting.

Table 1: The average highest returns over 10 instances on the MuJoCo control tasks at 400K and 1M time steps. ± captures the standard deviation over trials. The best score is highlighted by `cyan` and the second best score is highlighted by `orange.`

| Algorithm | Time step | Walker2d | HalfCheetah | Ant | Humanoid | Hopper |
|---|---|---|---|---|---|---|
| TD3 | 400K | 2636±933 | 8229±757 | 3297±1084 | 1384±282 | 2876±859 |
| | 1M | 4198±516 | 10560±675 | 4617±1287 | 5308±105 | 3387±137 |
| SAC | 400K | 3122±156 | 8945±1368 | 3893±569 | 2268±905 | 3276±86 |
| | 1M | 3921±163 | 11729±258 | 5956±2209 | 5498±131 | 3422±87, |
| TQC | 400K | 4994±397 | 9644±1006 | 3307±939 | 4061±703 | 3534±91 |
| | 1M | 5895±552 | 13431±561 | 5258±1165 | 6140±426 | 3602±117 |
| TD3+OFE | 400K | 4329±550 | 11508±635 | 6406±549 | 5193±797 | 3471±45 |
| | 1M | 4574±551 | 14759±696 | 7246±497 | 7262±209 | 3616±28 |
| TD7 | 400K | 5787±444 | 15625±559 | 7305±197 | 5823±231 | 3440±92 |
| | 1M | 6354±209 | 17343±359 | 8346±291 | 7405±236 | 3757±214 |
| HA3C | 400K | 6441±366 | 16652±323 | 7838±138 | 6099±305 | 3783±153 |
| | 1M | 7143±456 | 18108±294 | 8687±128 | 8584±273 | 4143±170 |

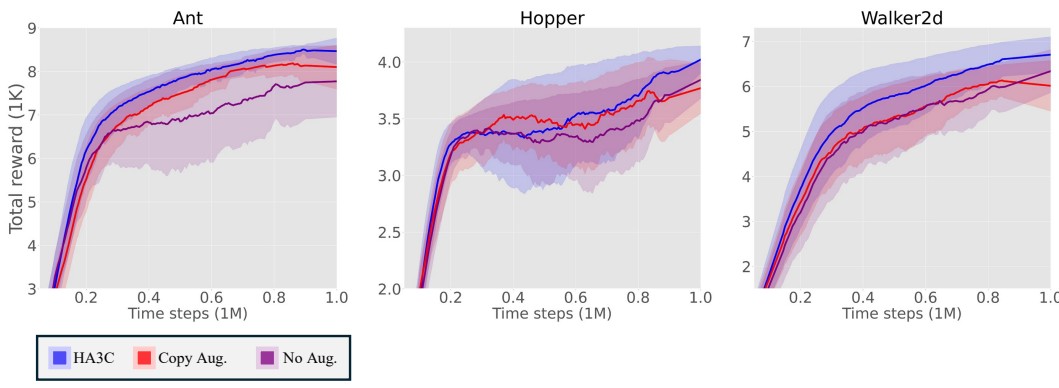

Figure 7: Learning curves of the ablation study on the MuJoCo benchmark. The shaded area captures a 90% confidence interval around the average performance.

As we can see from Fig. 7, HA3C significantly outperforms the compared algorithms in terms of both sample efficiency and performance on Ant and Walker2d. HA3C also significantly outperforms the compared algorithms in final performance on Hopper. This phenomenon illustrates that historical augmentation is the real source for improving sample efficiency.

## 5.3 PARAMETER SENSITIVITY ANALYSIS

In Fig. 8, we analyze the sensitivities of two important parameters, $k$ and $N$, on Ant. $k$ is the length of the historical state trajectory and $N$ is the number of dimensions of compressed historical trajectories. Both of the above parameters are not used in the previous representation-based RL algorithms. $k$ is set from $\{6, 12, 18, 24\}$ and $N$ is set from $\{8, 16, 64, 256\}$.

As we can see, HA3C is a little sensitive to $k$ and $N$. When $k \leq 12$ and $N \leq 16$, our historical augmentation will significantly improve the sample efficiency. When $N = 256$, the historical information cannot improve neither sample efficiency nor final performance. This phenomenon illustrates that compressing the historical trajectories into a low-dimensional space is the key to effectively utilize the historical information in MDP tasks.

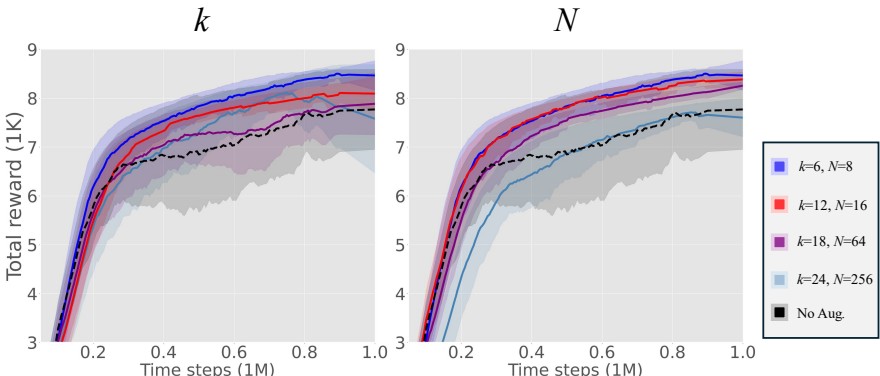

Figure 8: Learning curves of the parameter sensitivity analysis on the MuJoCo benchmark. The shaded area captures a 90% confidence interval around the average performance.

## 5.4 RUNNING TIME

To understand the computational cost of HA3C, we compare the running times of different algorithms with identical computational resources in HalfCheetah control task. The result is shown in Fig. 9. As we can see, the computational cost of HA3C is less than the computational costs of TD3+OFE and TQC but is more than the computational costs of TD3, SAC, and TD7.

## 6 CONCLUSION

Under the Markov assumption of MDPs, the probability distributions of the next state and reward depend only on the current state and action. Therefore, given a finite $Q$-table, we can find the optimal policy in an MDP by a heuristic algorithm which only considers single-step transitions. Different from the heuristic algorithm, DRL algorithms need to approximate the causal functions by learning the causal relationships in MDPs. In this case, DRL is often faced with sample inefficiency from complex causal relationships, as a more complex causal function requires neural networks to approximate with more parameters, samples, and time consumption.

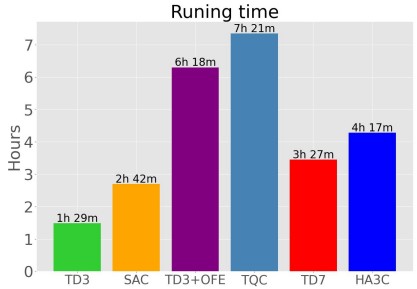

Figure 9: Running times of different algorithms for 1M time steps.

This paper addresses the above problem by augmenting the current state with historical information. We believe that historical augmentation can simplify the causal relationships of state transitions by its inherent contextual information and increasing the search space of the causal functions. Therefore, we focus on optimizing a history-dependent stationary policy in MDPs and propose a new RL algorithm, HA3C. The main idea of HA3C is to learn the state representations by compressing the high-dimensional historical trajectories into a low-dimensional space. In this way, we can extract the simple causal relationships from historical trajectories and avoid the overfitting caused by high-dimensional historical data. Our experiment demonstrates the superior performance of HA3C over five state-of-the-art RL algorithms on MuJoCo control tasks.

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

# A  DIFFERENT POLICIES

Time-related policies can be History-dependent ($H$) or $k$-order Markov ($M_k$) (Derman et al., 2020; Puterman, 2014). Denote $\mathcal{H}_t$ as the set of possible histories up to time step $t$. A history-dependent policy $\pi = \{d_t | t = 0, 1, ...\}$ at $t$ maps histories to actions as $d_t : \mathcal{H}_t \mapsto \mathcal{A}$. A $k$-order Markov policy $\pi = \{d_t | t = 0, 1, ...\}$ at $t$ maps $k$-order state transition trajectories to actions as $d_t : \mathcal{S}_{k,t} \mapsto \mathcal{A}$. A $k$-order stationary ($S_k$) policy is unrelated to time as $\pi : \mathcal{S}_{k,:} \mapsto \mathcal{A}$. In general, a randomized ($R$) policy selects the actions by a probability distribution as $\pi(\boldsymbol{a}|*)$. $\pi$ is a deterministic ($D$) policy if and only if $\pi(\boldsymbol{a}|*) \in \{0, 1\}$. Based on the above analysis, we can obtain History-dependent Random ($HR$) policies, History-dependent Deterministic ($HD$) policies, $k$-order Markov Random ($M_k R$) policies, $k$-order Markov Deterministic ($M_k D$) policies, $k$-order Stationary Random ($S_k R$) policies, and $k$-order Stationary Deterministic ($S_k D$) policies.

The above policies are summarized in Table 2. The relationships among them are demonstrated in Fig. 10. It is noteworthy that sometimes historical actions will be considered in decision-making. In this case, without loss of generality, a historical state $\boldsymbol{s}_{i|i \leq t-1}$ can be updated by $\boldsymbol{s}_i \leftarrow \boldsymbol{s}_i \cup \boldsymbol{a}_i$.

Table 2: Different types of policies.

| Policy | Abbreviation | Action |
|---|---|---|
| History-dependent Random | $HR$ | $\boldsymbol{a}_t \sim d_t(\boldsymbol{s}_{0,t}), d_t \in \pi$ |
| History-dependent Deterministic | $HD$ | $\boldsymbol{a}_t = d_t(\boldsymbol{s}_{0,t}), d_t \in \pi$ |
| $k$-order Markov Random | $M_k R$ | $\boldsymbol{a}_t \sim d_t(\boldsymbol{s}_{k-t+1,t}), d_t \in \pi$ |
| $k$-order Markov Deterministic | $M_k D$ | $\boldsymbol{a}_t = d_t(\boldsymbol{s}_{k-t+1,t}), d_t \in \pi$ |
| $k$-order Stationary Random | $S_k R$ | $\boldsymbol{a}_t \sim \pi(\boldsymbol{s}_{k-t+1,t})$ |
| $k$-order Stationary Deterministic | $S_k D$ | $\boldsymbol{a}_t = \pi(\boldsymbol{s}_{k-t+1,t})$ |

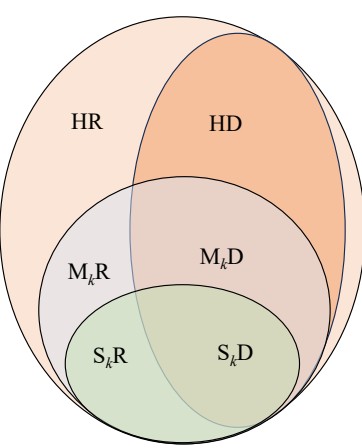

Figure 10: The relations among different policies.

# B  AN EXAMPLE OF IMPROVING SAMPLE EFFICIENCY IN MDPS BY HISTORICAL AUGMENTATION

Define a sequence as follows: 1) $|\beta_0| \neq 1$; 2) If $i > 1$, then $\beta_{i+1} = \beta_i^2$.

Based on the sequence above, we can define an MDP $\mathbb{M} = \langle \mathcal{S}, \mathcal{A}, R, \boldsymbol{P}, \gamma \rangle$. At time step $t$, state $\boldsymbol{s}_t = [\beta_t, \beta_{t+2}]^\top$ and action $\boldsymbol{a}_t$ is computed by a linear function $f(*)$ on state $\boldsymbol{s}_t$ or augmented state $\boldsymbol{s}_{k,t}$. Without considering historical information, reward $r_t$ is defined as

$$r_t = -|f(\boldsymbol{s}_t) - (\beta_t + \sqrt{\beta_{t+2}} + \beta_{t+2})| = -|\boldsymbol{w}\boldsymbol{s}_t + b - (\beta_t + \sqrt{\beta_{t+2}} + \beta_{t+2})|, \tag{6}$$

where $\boldsymbol{w}$ is a two-dimensional vector and $b$ is a constant. In transition model $\boldsymbol{P}$, $\boldsymbol{s}_0$ can be defined as $[\beta_0, \beta_2]^\top$ and $\boldsymbol{s}_{t+1}$ can be computed by $\boldsymbol{s}_t$ as

$$\boldsymbol{s}_{t+1} = [\beta_t^2, \beta_{t+2}^2]^\top = \boldsymbol{s}_t \odot \boldsymbol{s}_t, \tag{7}$$

where $\odot$ is Hadamard product. $\gamma = 0.99$.

From equation 6 and equation 7, it is easy to see that $\mathbb{M}$ satisfies the Markov assumption of MDPs. To maximize the discounted cumulative reward in $\mathbb{M}$, we should minimize

$$\arg\min_{\boldsymbol{w},b} ||f(\boldsymbol{s}_t) - (\beta_t + \sqrt{\beta_{t+2}} + \beta_{t+2})||_2 = \arg\min_{\boldsymbol{w},b} ||\boldsymbol{w}\boldsymbol{s}_t + b - (\beta_t^2 + \sqrt{\beta_{t+2}} + \beta_{t+2})||_2 \tag{8}$$

at each time step $t$. However, it is hard to minimize equation 8 by $f(\boldsymbol{s}_t)$, which is a linear model on $\boldsymbol{s}_t$.

The above problem can be solved by the historical augmentation of $\boldsymbol{s}_t$. When considering the historical augmentation of $\boldsymbol{s}_t$, $f(*)$ on $\boldsymbol{s}_{2,t}$ can be defined as

$$f(\boldsymbol{s}_{2,t}) = \boldsymbol{w}_0 \boldsymbol{s}_t + \boldsymbol{w}_1 \boldsymbol{s}_{t-1} + b.$$

Instead of minimizing equation 8, we can minimize

$$\arg\min_{\boldsymbol{w}_0, \boldsymbol{w}_1, b} ||f(\boldsymbol{s}_{2,t}) - (\beta_t + \sqrt{\beta_{t+2}} + \beta_{t+2})||_2$$
$$= \arg\min_{\boldsymbol{w}_0, \boldsymbol{w}_1, b} ||\boldsymbol{w}_0 \boldsymbol{s}_t + \boldsymbol{w}_1 \boldsymbol{s}_{t-1} + b - (\beta_t^2 + \sqrt{\beta_{t+2}} + \beta_{t+2})||_2.$$

Let $\boldsymbol{w}_0 = [1, 1]$, $\boldsymbol{w}_1 = [0, 1]$, and $b = 0$. From $\beta_{t+1} = \sqrt{\beta_{t+2}}$, we have

$$||\boldsymbol{w}_0 \boldsymbol{s}_t + \boldsymbol{w}_1 \boldsymbol{s}_{t-1} + b - (\beta_t + \sqrt{\beta_{t+2}} + \beta_{t+2})||_2$$
$$= ||\boldsymbol{w}_0 \boldsymbol{s}_t + \boldsymbol{w}_1 \boldsymbol{s}_{t-1} + b - (\beta_t + \beta_{t+1} + \beta_{t+2})||_2$$
$$= ||([1,1][\beta_t, \beta_{t+2}]^\top + [0,1][\beta_{t-1}, \beta_{t+1}]^\top - (\beta_t + \beta_{t+1} + \beta_{t+2})||_2$$
$$= 0.$$

In this case, the cumulative reward in $\mathbb{M}$ can be maximized.

# C  CONNECTED TO RELATED WORK

## C.1  CONNECTED TO HMDPS

In HMDPs, the probability distributions of the reward and next state depend not only on the current state and action but also on the historical states and actions. For a $k$-order HMDPs, we have

$$P\{\boldsymbol{s}_{t+1} = \boldsymbol{s}', r_t = r | \boldsymbol{s}_0, \boldsymbol{a}_0, r_0, ..., \boldsymbol{s}_t, \boldsymbol{a}_t\} = P\{\boldsymbol{s}_{t+1} = \boldsymbol{s}', r_t = r | \boldsymbol{s}_{t-k+1}, \boldsymbol{a}_{t-k+1}, .., \boldsymbol{s}_t, \boldsymbol{a}_t\}.$$

The causal diagram of HMDP is presented in Fig. 11(a). Our approach optimizes the policy by a simplified HMDP model in which the probability distributions of the reward and next state depend on the current state-action pair and compressed historical trajectory as

$$P\{\boldsymbol{s}_{t+1} = \boldsymbol{s}', r_t = r | \boldsymbol{s}_0, \boldsymbol{a}_0, r_0, ..., \boldsymbol{s}_t, \boldsymbol{a}_t\} = P\{\boldsymbol{s}_{t+1} = \boldsymbol{s}', r_t = r | DR(\boldsymbol{s}_{t-1,k-1}), .., \boldsymbol{s}_t, \boldsymbol{a}_t\}.$$

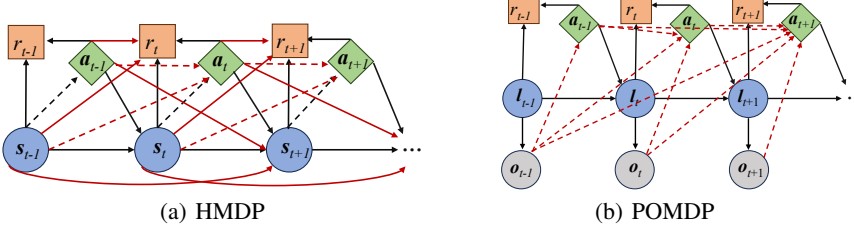

(a) HMDP            (b) POMDP

Figure 11: Causal diagram of HMDPs and POMDPs.

## C.2 CONNECTED TO POMDPS

In POMDPs, the states are partially observable. Define the partially observable state at time step $t$ as $l_t$ and the observable part of $l_t$ as $o_t$. The causal diagram of POMDPs is shown in Fig. 11(b). Under the faithfulness assumption, $o_t$ and $o_{t+k}$ are mutually dependent conditional on $\forall k > 1$, $\{o_i, a_i\}_{t<i<t+k}$ (Kalisch & Bühlman, 2007). Therefore, in a POMDP, the optimal policy $\pi$ at time step $t$ should consider not only $o_t$ but also the historical information $\{o_i, a_i\}_{0 \le i < t}$. When $k$ is large, long-length rollout estimation is needed in POMDPs.

RL algorithms of world models, such as Dream, model the sequential decision-making as a POMDP (Ha & Schmidhuber, 2018; Hafner et al., 2019a). They usually encode the historical information at $t$ by an encoder $f^t$ to construct $s_{t+1}$ as

$$s_{t+1} = f^t(o_t, a_t, ..., f^1(o_1, a_1, f^0(o_0, a_0))).$$

When $t$ is large, some partially observable states will be encoded many times, leading to the loss of some important discriminative information.

Compared with POMDP-based RL algorithms, our HA3C can better adjust the considered steps in history according to the actual task and thus effectively find the optimal policy in history-based sequential decision-making.

## C.3 CONNECTED TO STATE ABSTRACTION

State abstraction aims to reduce ground MDPs with large state spaces to abstract MDPs with smaller state spaces by aggregating states according to some notion of equality or similarity (Bartlett, 1966). Through abstraction, intelligent agents may need to consider only the salient distinguishing information of their environments. Given an abstraction function as $F : \mathcal{S} \to \overline{\mathcal{S}}$, we can define the abstract version of MDP $\mathbb{M}$ as $\overline{\mathbb{M}} = \langle \overline{\mathcal{S}}, \mathcal{A}, \overline{R}, \overline{P}, \gamma \rangle$. A $Q$-irrelevance abstraction function $F^Q$ is that for any action $a$, $F^Q(s) = F^Q(s')$ implies $Q(s, a) = Q(s', a)$. Then we have the following theorem.

**Theorem C.1.** *Define an MDP as $\mathbb{M}_k = \langle \mathcal{S}_{k,:}, \mathcal{A}, R, P_k, \gamma \rangle$. Under the conditions 1), 2), and 3) in Theorem 4.2, encoder $f$ is a $Q$-irrelevance abstraction on $s_{k,:}$.*

Theorem C.1 illustrates that our representation learning can be seen as the $Q$-irrelevance abstraction of the historically augmented states. The proof of this theorem is given in Appendix D.

# D THEORETICAL ANALYSIS

## D.1 PROOF OF THEOREM 4.1

Now we give the proof to Theorem 4.1. The different types of policies in this proof are summarized in Table 2. The relationships between these policies are shown in Fig. 10.

Based on the Markov assumption of MDPs, we have

$$P\{s_{t+1} = s', r_t = r | s_0, a_0, r_0, ..., s_t, a_t\} \tag{9}$$
$$= P\{s_{t+1} = s', r_t = r | s_{t-k+1,t}, a_t\}$$
$$= P\{s_{t+1} = s', r_t = r | s_t, a_t\}.$$

For any $\pi \in HR$, we can define $V_\pi(\boldsymbol{h}_t)$ by

$$V^\pi(\boldsymbol{h}_t) = \mathbb{E}^\pi \left[ \sum_{i=t}^{+\infty} \gamma^i R(\boldsymbol{h}_{t+i}, \boldsymbol{a}_{t+i}) \right].$$

From Fig. 10, we have $S_k D \in M_k D \in M_k R \in HR$. In view of equation 9, we see for all $t$ that

$$\sup_{\pi \in HR} V^\pi(\boldsymbol{h}_t) = \sup_{\pi \in S_k D} V^\pi(\boldsymbol{s}_{k,t}).$$

First, for all $t$, we demonstrate that

$$\sup_{\pi \in HR} V^\pi(\boldsymbol{h}_t) = \sup_{\pi \in M_k R} V^\pi(\boldsymbol{s}_{k,t}). \tag{10}$$

This is a direct result of Theorem D.1. The proof of this theorem is presented in D.1.1.

**Theorem D.1.** *Let $\pi = \{d_t | t = 0, 1, ...\} \in HR$. Then $\forall \boldsymbol{s}_{k,:} \in \mathcal{S}_{k,:}$, based on equation 9, there exists a policy $\pi' = \{d_t' | t = 0, 1, ...\} \in M_k R$ satisfying*

$$p^\pi(\boldsymbol{a}_{t+i} = \boldsymbol{a}', \boldsymbol{s}_{k,t+i} = \boldsymbol{s}_{k,:}' | \boldsymbol{s}_{k,t} = \boldsymbol{s}_{k,:}) = p^{\pi'}(\boldsymbol{a}_{t+i} = \boldsymbol{a}', \boldsymbol{s}_{k,t+i} = \boldsymbol{s}_{k,:}' | \boldsymbol{s}_{k,t} = \boldsymbol{s}_{k,:}),$$

*where $p^\pi(*)$ denotes the probability of $*$ provided that the agent follows policy $\pi$.*

Then Theorem D.2 illustrates that the value functions of $\pi \in M_k D$ and $\pi \in M_k R$ have the same upper bound. The proof of this theorem is demonstrated in D.1.2.

**Theorem D.2.** *If a bounded function $V$ on $\mathcal{S}_{k,:}$ satisfies the optimal Bellman equation that*

$$V(\boldsymbol{s}_{k,t}) = \sup_{\boldsymbol{a} \in \mathcal{A}} \left\{ R(\boldsymbol{s}_{k,t}, \boldsymbol{a}) + \gamma \int_{\mathcal{S}_{k,:}} V(\boldsymbol{s}_{k,t+1} | \boldsymbol{s}_{t+1} = \boldsymbol{s}') p(\boldsymbol{s}' | \boldsymbol{s}_{k,t}, \boldsymbol{a}) d\boldsymbol{s}_{k,:}' \right\},$$

*then*

$$\sup_{\pi \in M_k D} V^\pi(\boldsymbol{s}_{k,t}) = \sup_{\pi \in M_k R} V^\pi(\boldsymbol{s}_{k,t}).$$

Finally, based on equation 9, for all $\boldsymbol{s}_{k,:} \in \mathcal{S}_{k,:}$, if $\boldsymbol{s}_{k,t} = \boldsymbol{s}_{k,:}$, then

$$\sup_{\boldsymbol{a} \in \mathcal{A}} V(\boldsymbol{s}_{k,t}) = \sup_{\boldsymbol{a} \in \mathcal{A}} V(\boldsymbol{s}_{k,:}). \tag{11}$$

Let $\boldsymbol{a} = \pi(\boldsymbol{s}_{k,:})$, where $\pi \in S_k D$. It follows that

$$\sup_{\pi \in S_k D} V^\pi(\boldsymbol{s}_{k,:}) = \sup_{\pi \in M_k D} V^\pi(\boldsymbol{s}_{k,t}). \tag{12}$$

Under equation 10, equation 11 and equation 12, $\forall t$, if $\boldsymbol{s}_{k,t} = \boldsymbol{s}_{k,:}$, then

$$\sup_{\pi \in HR} V^\pi(\boldsymbol{h}_t) = \sup_{\pi \in M_k R} V^\pi(\boldsymbol{s}_{k,t}) = \sup_{\pi \in M_k D} V^\pi(\boldsymbol{s}_{k,t}) = \sup_{\pi \in S_k D} V^\pi(\boldsymbol{s}_{k,:}).$$

### D.1.1 PROOF OF THEOREM D.1

We assume that Theorem D.1 holds for $i = 1, 2, 3, ..., n - 1$. Given a policy $\pi \in HR$, based on equation 9, we see that there exists a policy $\pi' \in M_k R$ satisfying

$$p^\pi(\boldsymbol{s}_{k,t+i} = \boldsymbol{s}_{k,:}'' | \boldsymbol{s}_{k,t} = \boldsymbol{s}_{k,:})$$

$$= \int_{\mathcal{S}_{k,:}} \int_{\mathcal{A}} p^\pi(\boldsymbol{s}_{k,t+i-1} = \boldsymbol{s}_{k,:}', \boldsymbol{a}_{t+i-1} = \boldsymbol{a}' | \boldsymbol{s}_{k,t} = \boldsymbol{s}_{k,:}) p(\boldsymbol{s}'' | \boldsymbol{s}_{k,:}', \boldsymbol{a}') d\boldsymbol{a}' d\boldsymbol{s}_{k,:}'$$

$$= \int_{\mathcal{S}_{k,:}} \int_{\mathcal{A}} p^{\pi'}(\boldsymbol{s}_{k,t+i-1} = \boldsymbol{s}_{k,:}', \boldsymbol{a}_{t+i-1} = \boldsymbol{a}' | \boldsymbol{s}_{k,t} = \boldsymbol{s}_{k,:}) p(\boldsymbol{s}'' | \boldsymbol{s}_{k,:}', \boldsymbol{a}') d\boldsymbol{a}' d\boldsymbol{s}_{k,:}'$$

$$= p^{\pi'}(\boldsymbol{s}_{k,t+i} = \boldsymbol{s}_{k,:}'' | \boldsymbol{s}_{k,t} = \boldsymbol{s}_{k,:}).$$

The above equality follows from the induction hypothesis. The $\pi'$ also can satisfy

$$p^{\pi'}(\boldsymbol{a}_{t+i} = \boldsymbol{a}' | \boldsymbol{s}_{k,t+i} = \boldsymbol{s}_{k,:}') = p^\pi(\boldsymbol{a}_{t+i} = \boldsymbol{a}' | \boldsymbol{s}_{k,t+i} = \boldsymbol{s}_{k,:}').$$

Therefore,

$$p^{\pi'}(\boldsymbol{a}_{t+i} = \boldsymbol{a}', \boldsymbol{s}_{k,t+i} = \boldsymbol{s}_{k,:}' | \boldsymbol{s}_{k,t} = \boldsymbol{s}_{k,:})$$

$$= p^{\pi'}(\boldsymbol{a}_{t+i} = \boldsymbol{a}' | \boldsymbol{s}_{k,t+i} = \boldsymbol{s}_{k,:}') p^{\pi'}(\boldsymbol{s}_{k,t+i} = \boldsymbol{s}_{k,:}' | \boldsymbol{s}_{k,t} = \boldsymbol{s}_{k,:})$$

$$= p^\pi(\boldsymbol{a}_{t+i} = \boldsymbol{a}' | \boldsymbol{s}_{k,t+i} = \boldsymbol{s}_{k,:}') p^\pi(\boldsymbol{s}_{k,t+i} = \boldsymbol{s}_{k,:}' | \boldsymbol{s}_{k,t} = \boldsymbol{s}_{k,:})$$

$$= p^\pi(\boldsymbol{a}_{t+i} = \boldsymbol{a}', \boldsymbol{s}_{k,t+i} = \boldsymbol{s}_{k,:}' | \boldsymbol{s}_{k,t} = \boldsymbol{s}_{k,:}).$$

### D.1.2 PROOF OF THEOREM D.2

In view of $M_k D \in M_k R$, we have

$$\sup_{\pi \in M_k D} V^\pi(\boldsymbol{s}_{k,t}) \leq \sup_{\pi \in M_k R} V^\pi(\boldsymbol{s}_{k,t}). \tag{13}$$

It follows that

$$\sup_{\boldsymbol{a} \in \mathcal{A}} \left\{ R(\boldsymbol{s}_{k,t}, \boldsymbol{a}) + \gamma \int_{\mathcal{S}_{k,:}} V(\boldsymbol{s}_{k,t+1}|\boldsymbol{s}_{k,t+1} = \boldsymbol{s}')p(\boldsymbol{s}'|\boldsymbol{s}_{k,t}, \boldsymbol{a}))d\boldsymbol{s}' \right\}$$

$$\geq \int_{\mathcal{A}} p(d_t(\boldsymbol{s}_{k,t}) = \boldsymbol{a}) \left[ R(\boldsymbol{s}_{k,t}, \boldsymbol{a}) + \gamma \int_{\mathcal{S}_{k,:}} V(\boldsymbol{s}_{k,t+1}|\boldsymbol{s}_{t+1} = \boldsymbol{s}')p(\boldsymbol{s}'|\boldsymbol{s}_{k,t}, \boldsymbol{a}))d\boldsymbol{s}'_{k,:} \right] d\boldsymbol{a},$$

where $d_t \in M_k R$. Thus

$$\sup_{\pi \in M_k D} V^\pi(\boldsymbol{s}_{k,t}) \geq \sup_{\pi \in M_k R} V^\pi(\boldsymbol{s}_{k,t}). \tag{14}$$

Combining equation 13 and equation 14, we have

$$\sup_{\pi \in M_k D} V^\pi(\boldsymbol{s}_{k,t}) = \sup_{\pi \in M_k R} V^\pi(\boldsymbol{s}_{k,t}).$$

### D.2 PROOF OF THEOREM 4.2

To prove Theorem 4.2, we give the proof of Theorem C.1 first. Under the condition 1) of Theorem 4.2, one sees that there are only two independent variables $\boldsymbol{s}_{k,:}$ and $\boldsymbol{a}$. Under the Markov assumption and the condition 2) of Theorem 4.2, we have

$$P\{\boldsymbol{s}_{k,t+1} = \boldsymbol{s}'_{k,:}|\boldsymbol{s}_0, \boldsymbol{a}_0, r_0, ..., \boldsymbol{s}_t, \boldsymbol{a}_t\} = P\{\boldsymbol{s}_{k,t+1} = \boldsymbol{s}'_{k,:}|\boldsymbol{z}^{\boldsymbol{s}_{k,t}}, \boldsymbol{a}_t\}. \tag{15}$$

Then, under the condition 3) of Theorem 4.2, we have

$$
\begin{aligned}
P\{\boldsymbol{z}^{\boldsymbol{s}_{k,t+1}} = \boldsymbol{z}^{\boldsymbol{s}'_{k,:}}|\boldsymbol{s}_{k,t}, \boldsymbol{a}_t\} &\doteq P\{\boldsymbol{z}^{\boldsymbol{s}_{k,t+1}} = \boldsymbol{z}^{\boldsymbol{s}'_{k,:}}|\boldsymbol{z}^{\boldsymbol{s}_{k,t}, \boldsymbol{a}_t}\} \\
&= P\{\boldsymbol{z}^{\boldsymbol{s}_{k,t+1}} = \boldsymbol{z}^{\boldsymbol{s}'_{k,:}}|g(f(\boldsymbol{s}_{k,t}), \boldsymbol{a}_t)\} \\
&= P\{\boldsymbol{z}^{\boldsymbol{s}_{k,t+1}} = \boldsymbol{z}^{\boldsymbol{s}'_{k,:}}|g(\boldsymbol{z}^{\boldsymbol{s}_{k,t}}, \boldsymbol{a}_t)\} \\
&= P\{\boldsymbol{z}^{\boldsymbol{s}_{k,t+1}} = \boldsymbol{z}^{\boldsymbol{s}'_{k,:}}|\boldsymbol{z}^{\boldsymbol{s}_{k,t}}, \boldsymbol{a}_t\}.
\end{aligned}
\tag{16}
$$

Define an MDP as $\mathbb{M}_k = \langle \mathcal{S}_{k,:}, \mathcal{A}, R, \boldsymbol{P}_k, \gamma \rangle$. From equation 15 and equation 16, we obtain

$$\boldsymbol{z}^{\boldsymbol{s}_{k,:}} = \boldsymbol{z}^{\boldsymbol{s}'_{k,:}} \rightarrow Q(\boldsymbol{s}_{k,:}, \boldsymbol{a}) = Q(\boldsymbol{s}'_{k,:}, \boldsymbol{a})$$

Because $\boldsymbol{z}^{\boldsymbol{s}_{k,:}} = f(\boldsymbol{s}_{k,:})$, we see that encoder $f$ is a $Q$-irrelevance abstraction on $\boldsymbol{s}_{k,:}$.

Define an abstracted MDP of $\mathbb{M}_k$ as $\overline{\mathbb{M}}_k = \langle \mathcal{Z}, \mathcal{A}, R, \boldsymbol{P}_k, \gamma \rangle$, where $\mathcal{Z}$ is the encoded space of $\mathcal{S}_{k,:}$. Operator $B_\mu$ can be written as

$$B_\mu \hat{Q}(\boldsymbol{z}^{\boldsymbol{s}_{k,:}}, \boldsymbol{a}) = R(\boldsymbol{z}^{\boldsymbol{s}_{k,:}}, \boldsymbol{a}) + \max_\mu \gamma \int_{\mathcal{Z}} \hat{Q}(\boldsymbol{z}^{\boldsymbol{s}_{k,:}}, \mu(\boldsymbol{z}^{\boldsymbol{s}_{k,:}}))p(\boldsymbol{z}^{\boldsymbol{s}'_{k,:}}|\boldsymbol{z}^{\boldsymbol{s}_{k,:}, \boldsymbol{a}})d\boldsymbol{z}^{\boldsymbol{s}'_{k,:}}.$$

Now we provide a proof (sketch) to Theorem 4.2. Since the optimality of $\mu$ follows from the optimal actions as well as their $Q$-values are preserved after abstraction, we see that $B$ is a contraction in the sup-norm and the optimal $Q$-function $\hat{Q}^*$ is the unique fixed point of $B$. Thus we can finally find the optimal policy $\mu^*$ by $B_\mu$ (Melo, 2001). When the agent estimates the optimal $Q$-function based on experience, we have the following update rule in each time step $T$ by Lemma D.3 (Jaakkola et al., 1993; Melo, 2001).

$$\hat{Q}_{t+1}(\boldsymbol{z}^{\boldsymbol{s}_{k,t}}, \boldsymbol{a}_t) = \hat{Q}_t(\boldsymbol{z}^{\boldsymbol{s}_{k,t}}, \boldsymbol{a}_t) + \alpha_t(r_t + \gamma \max_\mu \hat{Q}_t(\boldsymbol{z}^{\boldsymbol{s}_{k,t+1}}, \mu(\boldsymbol{z}^{\boldsymbol{s}_{k,t+1}})) - \hat{Q}_t(\boldsymbol{z}^{\boldsymbol{s}_{k,t}}, \boldsymbol{a}_t)).$$

$\hat{Q}_t$ converges to $Q^*$ as long as

$$\sum_{t=0}^{\infty} \alpha_t = \infty, \quad \sum_{t=0}^{\infty} \alpha_t^2 < \infty.$$

**Lemma D.3.** *The random process $\{\Delta_t\}$ taking values in $\mathbb{R}^n$ and defined as*

$$\Delta_{t+1}(\boldsymbol{y}) = (1 - \alpha_t)\Delta_t(\boldsymbol{y}) + \alpha_t F_t(\boldsymbol{y})$$

*converges to zero under the following assumptions:*
*1) $\sum_{t=0}^{\infty} \alpha_t = \infty$ and $\sum_{t=0}^{\infty} \alpha_t^2 < \infty$,*
*2) $\mathbb{E}[||F_t(\boldsymbol{y})|\mathcal{F}_t||_w] \leq \gamma||\Delta_t||_w$ with $\gamma < 1$, and*
*3) $\text{Var}[F_t(\boldsymbol{y})|\mathcal{F}_t] \leq C(1 + ||\Delta_t||_w^2)$ for $C > 0$,*
*where $\mathcal{F} = \{\Delta_t, \Delta_{t-1}, ..., F_{t-1}, ..., \alpha_{t-1}, ..., \}$ strands for the past at step $t$ and $|| * ||_w$ refers to some weighted maximum norm.*

### D.3 APPROXIMATION ERROR ANALYSIS

Define the value function in $\mathcal{Z}$ as $\hat{V}$. The bound of the approximation error between the transition probabilities in space $\mathcal{S}_{k,:}$, and $\mathcal{Z}$ based on the optimal value function $\hat{V}^*$ can be defined as (Müller, 1997)

$$\max_{\boldsymbol{s}_{k,:},\boldsymbol{a}} \left| \int_{\mathcal{S}_{k,:}} \hat{V}^*(\boldsymbol{z}^{\boldsymbol{s}'_{k,:}})p(\boldsymbol{s}'_{k,:}|\boldsymbol{s}_{k,:},\boldsymbol{a})d\boldsymbol{s}'_{k,:} - \int_{\mathcal{Z}} \hat{V}^*(\boldsymbol{z}^{\boldsymbol{s}'_{k,:}})p(\boldsymbol{z}^{\boldsymbol{s}'_{k,:}}|\boldsymbol{z}^{\boldsymbol{s}_{k,:}},\boldsymbol{a})d\boldsymbol{z}^{\boldsymbol{s}'_{k,:}} \right| = \delta.$$

Based on $\delta$, we analyze the approximation error in Theorem D.4.

**Theorem D.4.** *The worst-case difference between $V^\mu(\boldsymbol{z}^{\boldsymbol{s}_{k,:}})$ and optimal value function $V^*(\boldsymbol{s})$ is bounded as:*

$$||V^*(\boldsymbol{s}) - \hat{V}^*(\boldsymbol{z}^{\boldsymbol{s}_{k,:}})||_\infty \leq \frac{\gamma\delta}{1 - \gamma}.$$

We provide the proof to the above theorem as follows. Based on the Markov assumption of MDPs, we have

$$||V^*(\boldsymbol{s}) - \hat{V}^*(\boldsymbol{z}^{\boldsymbol{s}_{k,:}})||_\infty = ||V^*(\boldsymbol{s}_{k,:}) - \hat{V}^*(\boldsymbol{z}^{\boldsymbol{s}_{k,:}})||_\infty.$$

Now we prove that

$$||V^*(\boldsymbol{s}_{k,:}) - \hat{V}^*(\boldsymbol{z}^{\boldsymbol{s}_{k,:}})||_\infty \leq \frac{\gamma\delta}{1 - \gamma}. \tag{17}$$

In view of $R(\boldsymbol{s}, \boldsymbol{a}) = R(\boldsymbol{s}_{k,:}, \boldsymbol{a}) = R(\boldsymbol{z}^{\boldsymbol{s}_{k,:}}, \boldsymbol{a})$ in the value function approximation, we have

$$
\begin{aligned}
&||V^*(\boldsymbol{s}_{k,:}) - \hat{V}^*(\boldsymbol{z}^{\boldsymbol{s}_{k,:}})||_\infty \\
\leq\ & \max_{\boldsymbol{s}_{k,:},\boldsymbol{a}} ||Q^*(\boldsymbol{s}_{k,:}, \boldsymbol{a}) - \hat{Q}^*(\boldsymbol{z}^{\boldsymbol{s}_{k,:}}, \boldsymbol{a})|| \\
=\ & \max_{\boldsymbol{s}_{k,:},\boldsymbol{a}} \left| R(\boldsymbol{s}_{k,:}, \boldsymbol{a}) + \gamma \int_{\mathcal{S}_{k,:}} V^*(\boldsymbol{s}'_{k,:})p(\boldsymbol{s}'_{k,:}|\boldsymbol{s}_{k,:},\boldsymbol{a})d\boldsymbol{s}'_{k,:} \right. \\
&\ \left. - R(\boldsymbol{z}^{\boldsymbol{s}_{k,:}}, \boldsymbol{a}) - \gamma \int_{\mathcal{Z}} \hat{V}^*(\boldsymbol{z}^{\boldsymbol{s}'_{k,:}})p(\boldsymbol{z}^{\boldsymbol{s}'_{k,:}}|\boldsymbol{z}^{\boldsymbol{s}_{k,:}},\boldsymbol{a})d\boldsymbol{z}^{\boldsymbol{s}'_{k,:}} \right| \\
\leq\ & \gamma \max_{\boldsymbol{s}_{k,:},\boldsymbol{a}} \left| \int_{\mathcal{S}_{k,:}} V^*(\boldsymbol{s}'_{k,:})p(\boldsymbol{s}'_{k,:}|\boldsymbol{s}_{k,:},\boldsymbol{a})d\boldsymbol{s}'_{k,:} - \hat{V}^*(\boldsymbol{z}^{\boldsymbol{s}'_{k,:}})p(\boldsymbol{s}'_{k,:}|\boldsymbol{s}_{k,:},\boldsymbol{a})d\boldsymbol{s}'_{k,:} \right| \\
&+\ \gamma \max_{\boldsymbol{s}_{k,:},\boldsymbol{a}} \left| \int_{\mathcal{S}_{k,:}} \hat{V}^*(\boldsymbol{z}^{\boldsymbol{s}'_{k,:}})p(\boldsymbol{s}'_{k,:}|\boldsymbol{s}_{k,:},\boldsymbol{a})d\boldsymbol{s}'_{k,:} - \int_{\mathcal{Z}} \hat{V}^*(\boldsymbol{z}^{\boldsymbol{s}'_{k,:}})p(\boldsymbol{z}^{\boldsymbol{s}'_{k,:}}|\boldsymbol{z}^{\boldsymbol{s}_{k,:}},\boldsymbol{a})d\boldsymbol{z}^{\boldsymbol{s}'_{k,:}} \right| \\
\leq\ & \gamma \left( ||V^*(\boldsymbol{s}_{k,:}) - \hat{V}^*(\boldsymbol{z}^{\boldsymbol{s}_{k,:}})||_\infty + \delta \right).
\end{aligned}
$$

This proves equation 17. Thus Theorem D.4 holds.

## D.4 ANALYZING SAMPLE EFFICIENCY IN EXPLORATION AND EXPLOITATION

In this subsection, we illustrate the benefit of sample efficiency from history augmentation based on two facts:

1) Historical augmentation can improve exploration in DRL. The policy can generate different actions for different transition trajectories that end with the same state;

2) Historical augmentation can also improve exploitation in DRL. History augmentation may simplify the causal relationships between the state and the explored high-reward action, thus the policy network can effectively learn and then regenerate this action.

The detailed analysis of these two facts is as follows. In the previous DRL methods for MDPs, when the policy $\mu$ and $s_t = s$ are fixed, we can get only one action by

$$\boldsymbol{a}_t = \mu(\boldsymbol{s}_t), \quad \mu \in S_1D.$$

However, based on our history-based policy

$$\boldsymbol{a}_t = \mu(\boldsymbol{s}_{k,t}), \quad \mu \in S_kD|_{k \geq 2}.$$

$\boldsymbol{a}_t$ can be changed by the change of the $\boldsymbol{s}_{k-1,t-1}$. We define the set of possible actions from policy $\mu \in S_kD$ at state $\boldsymbol{s}$ as $\boldsymbol{A}_\mu^s$ and the set of possible $k$-order trajectories end with state $\boldsymbol{s}$ as $\boldsymbol{S}_k^s$. As we can see, $|\boldsymbol{A}_\mu^s| \leq |\boldsymbol{S}_k^s|$.

Fig. 12 is the causal diagram of regenerating a high-reward action with or without historical augmentation. For a policy network $\mu_\theta \in S_1D$ and $\boldsymbol{a} = \mu_\theta(\boldsymbol{s})$, we may get $\boldsymbol{a}^* = \boldsymbol{a} + \epsilon$ with $R(\boldsymbol{s}, \boldsymbol{a}^*) > R(\boldsymbol{s}, \boldsymbol{a})$. However, it may be hard to regenerate $\boldsymbol{a}^*$ by the policy network $\mu_\theta(\boldsymbol{s})$ because the noise $\epsilon$ is independent of parameter $\theta$. Fortunately, the causal relationship between $\boldsymbol{s}_{k,t}|_{k \geq 2}$ and $\boldsymbol{a}^*$ may be simpler than the causal relationship between $\boldsymbol{s}_t$ and $\boldsymbol{a}^*$ (See the example in Appendix B). In this case, we can effectively learn the policy $\mu_\theta \in S_kD$ to regenerate the $\boldsymbol{a}^*$ at state $\boldsymbol{s}$ by $\boldsymbol{a}^* = \mu_\theta(\boldsymbol{s}_{k,t})$ (See the example in Fig. 5).

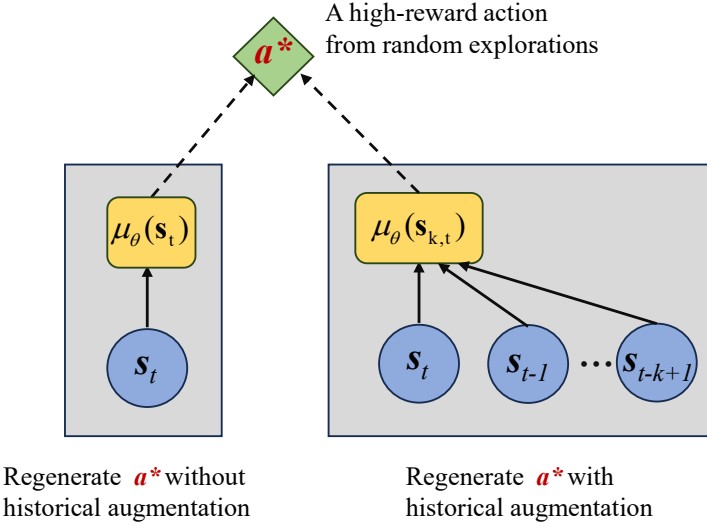

Figure 12: The causal diagram of regenerating a high-reward action with or without historical augmentation. The dashed lines indicate the information needed in the optimization.

# E EXPERIMENTAL SETTING

All experiments are run on a single Nvidia 3090 GPU and AMD 5900X CPU. We use the following software versions:
• Python 3.9.12
• Pytorch 2.0.0 (Paszke et al., 2019)
• CUDA 12.2
• Gymnasium 0.29.1 (Brockman et al., 2016)
• MuJoCo 3.2.3 (Todorov et al., 2012)

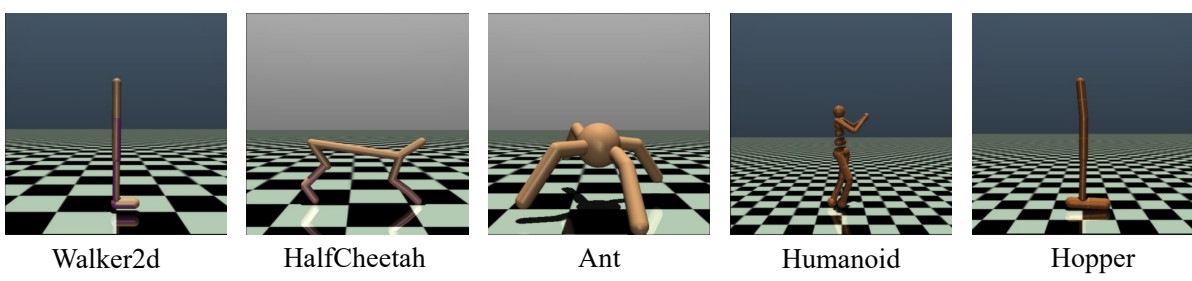

| Walker2d | HalfCheetah | Ant | Humanoid | Hopper |

Figure 13: The environments in our experiments.

The environments in our experiment are shown in Fig. 13 and detailed as follows:
1) Walker2d aims to walk in the forward direction as fast as possible.
2) HalfCheetah aims to run forward as fast as possible.
3) Ant aims to coordinate the four legs to move in the forward direction as fast as possible.
4) Humanoid aims to walk forward as fast as possible without falling over.
5) Hopper aims to make hops that move in the forward direction as fast as possible.

The compared RL algorithms in our experiment are detailed as follows.
• Online:
1) TD3 takes the minimum value between a pair of critic networks to address the overestimation of $Q$-value and reduces per-update error by delaying policy updates (Fujimoto et al., 2018).
2) SAC is an actor-critic algorithm based on the maximum entropy approach. The objective encourages policy stochasticity by augmenting the reward with the entropy at each step (Haarnoja et al., 2018).
3) OFE-TD3 increases the input dimensionality of the networks by representation learning to improve the sample efficiency of TD3 (Ota et al., 2020).
4) TQC addresses the overestimation of $Q$-value by the combination of the distributional representation of a critic, truncation of critic prediction, and ensembling of multiple critics (Kuznetsov et al., 2020).
5) TD7 is an effective DRL algorithm which combines TD3, state representation learning, checkpoints, prioritized experience replay, and a behaviour cloning term (only used for offline RL) (Fujimoto et al., 2023).

The hyper-parameters of HA3C are shown in Table 3. For Hopper, $\gamma$ is set as 0.992. Network architecture details are described in Pseudocode 1-3. The optimizer of our networks is Adam Kingma (2015).

Table 3: Hyper-parameters.

| Parameter | Value | Brief explanation |
|---|---|---|
| Start-timesteps | 25000 | Time steps of the initial random policy is used |
| Batch-size | 512 | Batch size for both actor and critic |
| $t_{pol}$ | 2 | Policy update frequency |
| $t_{tar}$ | 250 | Target update rate |
| $t_{ear}$ | 1 | Early assessment episodes for checkpoint operation |
| $t_{lat}$ | 3 | Late assessment episodes for checkpoint operation |
| $T_{ear}$ | 750K | Early time steps for checkpoint operation |
| $\sigma_e$ | 0.1 | Std of exploration noise |
| $\sigma_T$ | 0.005 | Std of target policy noise |
| $c$ | (-0.11,0.11) | Target policy noise clipping |
| $k$ | 6 | The length of the considering state sequences |
| $\gamma$ | 0.99 | Discount factor |
| $l_e$ | 0.0006 | The learning rate of the encoder network |
| $l_p$ | 0.0003 | The learning rate of the actor-network |
| $l_Q$ | 0.0003 | The learning rate of the network of the $Q$-functions |
| $\alpha$ | 0.25 | Controlling the amount of prioritization in LAP |
| $P_m$ | 1.1 | Minimum priority in LAP |

---

**Pseudocode 1: Critic network Details**

**Critic network:**
L1 = Linear(state-dim + action-dim, 256)
L2 = Linear($z^s$-dim * 2 + 256, 256)
L3 = Linear(256, 256)
L4 = Linear(256, 1)
**Critic forward pass:**
$x$ = Concatenate($[s_t, a_t]$)
$x$ = AvgL1Norm(L1($x$))
$x$ = Concatenate($[z^{s_{k,t},a_t}, z^{s_{k,t}}, x]$)
$x$ = Elu(L2($x$))
$x$ = Elu(L3($x$))
$\tau(s_{k,t}, a_t)$ = L4($x$)

---

**Pseudocode 2: Actor network Details**

**Actor network:**
L1 = Linear(state-dim, 256)
L2 = Linear($z^s$-dim + 256, 256)
L3 = Linear(256, 256)
L4 = Linear(256, action-dim)
**Actor forward pass:**
$x$ = AvgL1Norm(L1($s_t$))
$x$ = Concatenate($[z^{s_{k,t}}, x]$)
$x$ = ReLU(l1($x$))
$x$ = ReLU(l2($x$))
$a_t$ = Tanh(l3($x$))

**Pseudocode 3: Encoder Details**

**State Encoder $f$ Network:**
Conv = Conv2d(kernel-num=64, kernel-size=(3, state-dim), stride=1)
Pool = MaxPool2d((1, 1))
L1 = Linear(64, 16)
L2 = Linear(state-dim, 256)
L3 = Linear(256+16, 256)
L4 = Linear(256, zs-dim)

**State Encoder $f$ Forward Pass:**
$x = \text{Conv}(s_{k-1,t-1})$
$x = \text{Pool}(x)$
$x = \text{Elu}(\text{L1}(x))$
$x = \text{AvgL1Norm}(x)$
$y = \text{Elu}(\text{L2}(s_t))$
$x = \text{Concatenate}([x, y])$
$x = \text{Elu}(\text{L3}(x))$
$z^{s_{k,t}} = \text{AvgL1Norm}(\text{L4}(x))$

---

**State-Action Encoder $g$ Network:**
L1 = Linear(action-dim + $z^s$-dim, 256)
L2 = Linear(256, 256)
L3 = Linear(256, $z^s$-dim)
**State-Action Encoder $g$ Forward Pass:**
$x = \text{Concatenate}([a_t, z^{s_{k,t}}])$
$x = \text{Elu}(\text{L1}(x))$
$x = \text{Elu}(\text{L2}(x))$
$z^{s_{k,t}, a_t} = \text{L3}(x)$

# F SUPPLEMENTARY EXPERIMENT

## F.1 BIPEDAWALKER EXPERIMENT

To illustrate the benefit of history augmentation for complex MDP tasks, we test HA3C and No Aug. (HA3C without historical augmentation) on BipedalWalker and BipedalWalker-hardcore tasks. In BipedalWalker a robot is trained to move forward with slightly uneven terrain. Compared with BipedalWalker, BipedalWalker-hardcore is a more complex task, where the above robot is trained to move forward with ladders, stumps, and pitfalls. Therefore, the causal relationships in the transitions of BipedalWalker-hardcore are more complex than those in the transitions of BipedalWalker. The environments and learning curves are shown in Fig. 14 and the numerical results are shown in Table 4.

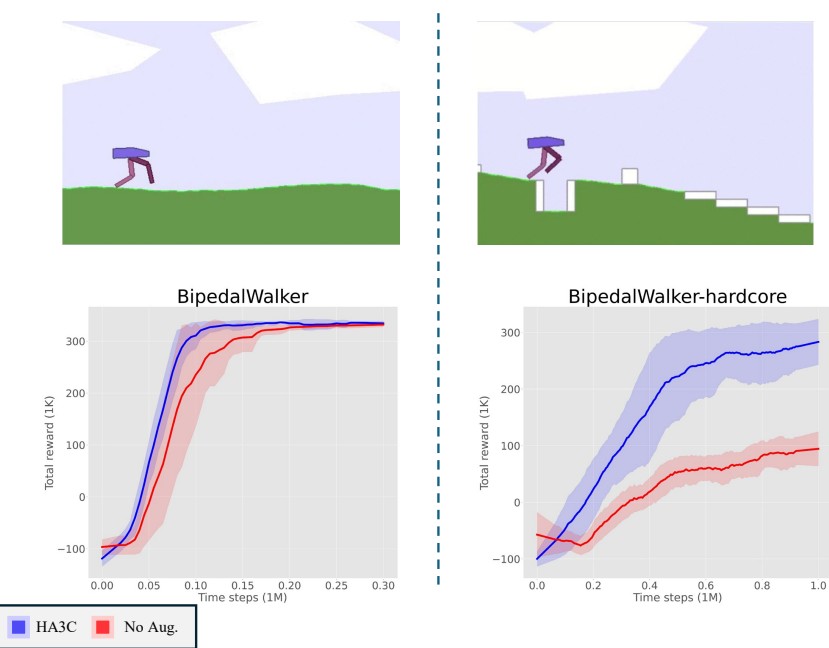

Figure 14: The environments and learning curves on BipedalWalker and BipedalWalker-hardcore tasks.

Table 4: The average highest returns of HA3C and No Aug. on BipedaWalker and BipedaWalker-hardcore tasks.

| Algorithm | BipedalWalke | BipedalWalker-hardcore |
|---|---|---|
| HA3C | 332 $\pm27$ | 316 $\pm19$ |
| No Aug. | 325 $\pm31$ | 171 $\pm21$ |

As we can see, although, both HA3C and No Aug. can get the high cumulative rewards in Bipedal-Walker, only HA3C can get the high cumulative rewards in BipedalWalker-hardcore. This is because by historical augmentation our HA3C can simplify the causal relationships in the transitions of BipedalWalker-hardcore.

## F.2 VISUALIZED RESULTS OF HA3C

Fig. 15 presents the visual results of the transitions in HA3C and No Aug. The collected states of each control task are mapped together by UMAP. The max learning step is $4 \times 10^5$ and each state is coloured by the reward of reaching it.

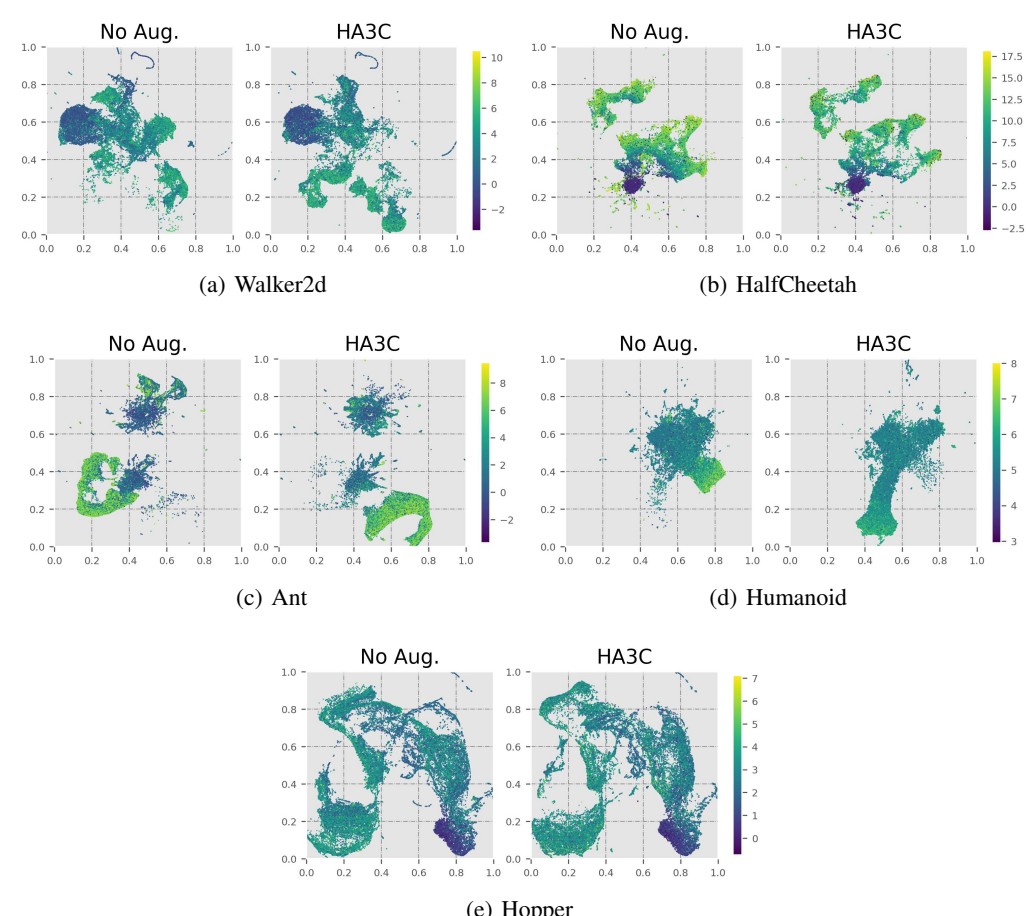

Figure 15: Visualized results of the explored states in No Aug. and HA3C.

As we can see, in Walker2d, Ant, and Humanoid, the high-reward states from HA3C are more than those from No Aug. This result illustrates that the sample efficiency of DRL can be effectively improved by learning the state representations with historical augmentation.

### F.3 DEEP MIND CONTROL SUITE EXPERIMENT

In this subsection, we evaluate our HA3C on five DMC tasks including ball_in_cup-catch, walker-run, quadruped-run, cheetah-run, and reacher-hard (Tassa et al., 2018). The compared algorithms are TD3 (Fujimoto et al., 2018) and TD7 (Fujimoto et al., 2023). For all algorithms, each task runs 10 instances with $10^6$ time steps with different random seeds. In each instance, the evaluation is performed every 5000 time steps. Some parameters are changed as follows. For quadruped-run, $l_e$ is set as 0.0006, $\sigma_T$ is set as 0.06, and $c$ is set as $(-0.12, 0.12)$. For other tasks, $l_e$ is set as 0.0005 and $c$ is set as $(-0.1, 0.1)$. The learning curves are shown in Fig. 18 and the numerical results at 300K time step and 1M time step are shown in Table 5.

As we can see, in most cases, HA3C has higher cumulative rewards than the compared algorithms. For walker-run, quadruped-run, and reacher-hard, HA3C outperforms the compared algorithms in terms of both the early performance and the final performance. For ball_in_cup-catch and cheetah-run, HA3C outperforms all of the compared algorithms in the final performance but the average return of HA3C is lower than the average return of TD7 in the early performance.

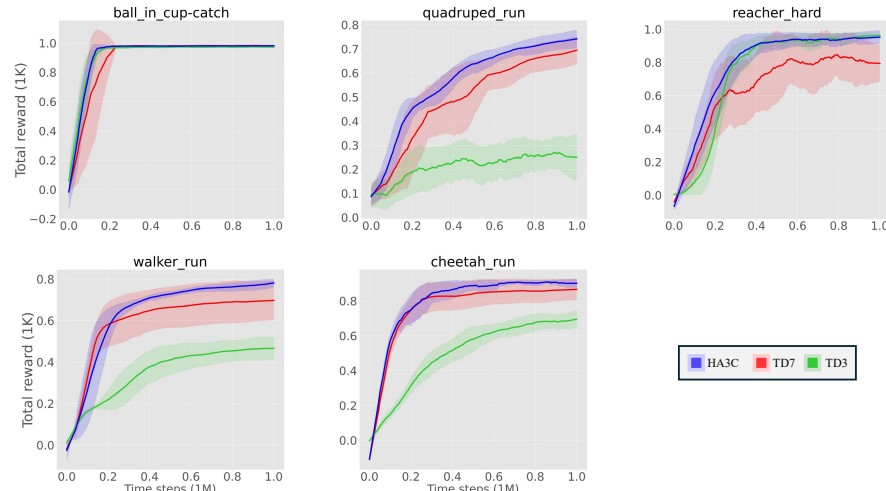

Figure 16: Learning curves of different RL algorithms on the deep mind control suite tasks.

Table 5: The average highest returns over 10 instances on the deep mind control suite tasks at 400K and 1M time steps.

| Algorithm | Time step | ball_in_cup-catch | walker-run | quadruped-run | cheetah-run | reacher-hard |
|-----------|-----------|-------------------|------------|---------------|-------------|--------------|
| TD3 | 400K | 981±2 | 387±71 | 331±65 | 550±76 | 971±3 |
| | 1M | 985±1 | 481±54 | 444±22 | 729±39 | 979±1 |
| TD7 | 400K | 990±2 | 654±96 | 531±69 | 836±75 | 879±91 |
| | 1M | 991±1 | 706±95 | 703±54 | 868±56 | 979±5 |
| HA3C | 400K | 989±2 | 713±41 | 598±36 | 834±108 | 976±5 |
| | 1M | 992±1 | 789±19 | 758±24 | 916±5 | 985±5 |

## F.4 LONGER TRAINING RUNS

In this section, we compare our HA3C with TD7 on five Mujoco control tasks with 3M training steps. The learning curves are shown in Fig. 17 and the numerical results are shown in Table 6.

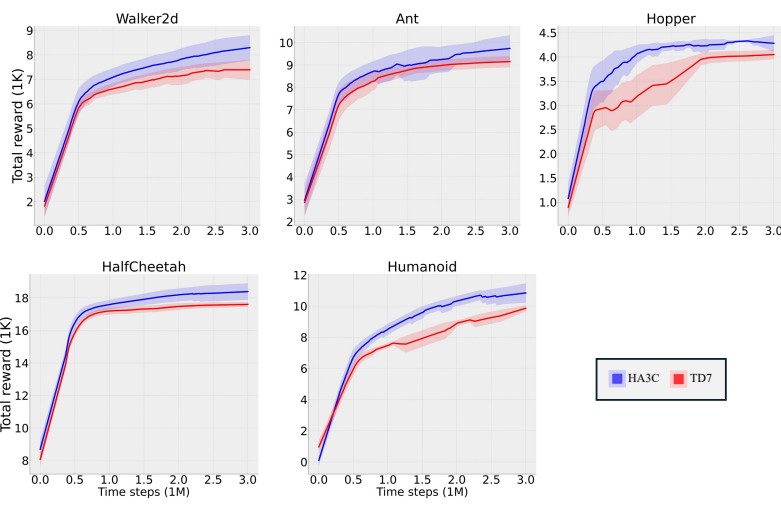

Figure 17: Learning curves of HA3C and TD7 on the Mujoco control tasks.

Table 6: The average highest returns over 10 instances of HA3C and TD7 at 3M time steps. $\pm$ captures the standard deviation over trials.

| Algorithm | Walker2d | HalfCheetah | Ant | Humanoid | Hopper |
|---|---|---|---|---|---|
| TD7 | $7570\pm321$ | $17787\pm286$ | $9225\pm450$ | $9850\pm226$ | $4049\pm156$ |
| HA3C | $8463\pm829$ | $18687\pm683$ | $9794\pm891$ | $11381\pm344$ | $4413\pm59$ |

As we can see, HA3C outperforms TD7 on the five Mujoco control tasks. It is noteworthy that the cumulative rewards of HA3C are significantly higher than the cumulative rewards of TD7 on Walker2d, Humanoid, and Hopper.

### F.5 COMBINING HISTORICAL REPRESENTATION LEARNING WITH SAC

In this section, we combine our historical representation learning with SAC to construct HA3C-SAC method (Haarnoja et al., 2018). Then we evaluate HA3C-SAC on three MuJoCo control tasks including Walker2d, Humanoid, and Hopper. The compared methods includes the original SAC and SALE-SAC, which combines the representation learning with SAC without historical augmentation (Fujimoto et al., 2023). The learning curves are shown in Fig. 17 and the numerical results are shown in Table 7.

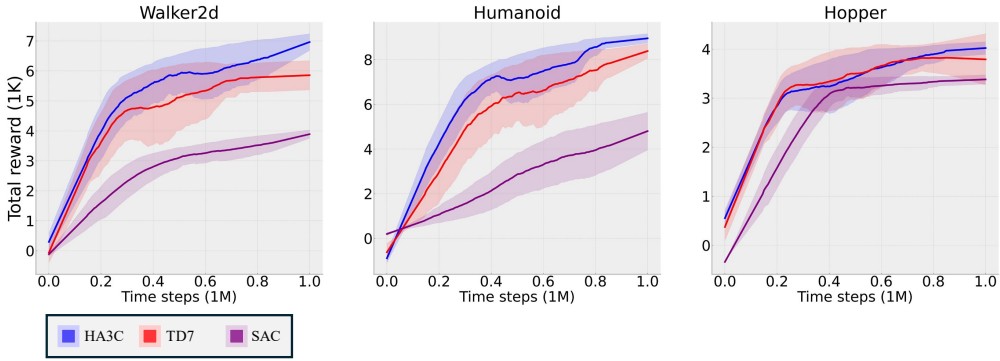

Figure 18: Learning curves of different RL algorithms on the deep mind control suite tasks.

Table 7: The average highest returns on Mujoco control tasks at 400K and 1M time steps.

| Algorithm | Time step | Walker2d | Humanoid | Hopper |
|---|---|---|---|---|
| SAC | 400K | $2843\pm148$ | $2268\pm905$ | $3195\pm33$ |
| | 1M | $3921\pm163$ | $5498\pm131$ | $3422\pm86$ |
| SALE-SAC | 400K | $5414\pm377$ | $6430\pm191$ | $3515\pm125$ |
| | 1M | $6021\pm492$ | $8368\pm330$ | $4038\pm126$ |
| HA3C-SAC | 400K | $5796\pm395$ | $7112\pm339$ | $3566\pm39$ |
| | 1M | $6950\pm623$ | $9047\pm238$ | $4131\pm48$ |

As we can see, HA3C-SAC outperforms SAC and SALE-SAC on the three Mujoco control tasks. The above results and the results Section 5.1 illustrate that our historical representation learning is robust to different algorithms and tasks.

