# OpenReview forum: "Beyond Markov Assumption: Improving Sample Efficiency in MDPs by Historical Augmentation"
_ICLR.cc/2025/Conference — Submitted to ICLR 2025_

### Official Review · Reviewer_CaX6 · 2024-10-27

**Soundness:** 2
**Presentation:** 3
**Contribution:** 2
**Rating:** 5
**Confidence:** 4

**Summary:**

This paper posits that, even when operating under the Markov assumption, it is beneficial for policy formulation to take into account not only current states but also historical information. This is based on the assumption that single-step state transitions might have complex causal relationships. Introducing historical data could potentially simplify these causal relationships, making them easier for neural networks to learn. On this basis, a novel Reinforcement Learning (RL) algorithm named HA3C is proposed, which has demonstrated superior performance over other advanced algorithms, such as TD3 and TD7, in five MuJoCo control tasks.

**Strengths:**

1. The perspective of causal relationships is interesting to understand why one should use historical information as additional inputs.
2. The theoretical formulation is precise and comprehensive.

**Weaknesses:**

1. The improvement of HA3C over baselines on the five Mujoco tasks appears to be subtle rather than significant.
2. It would be beneficial to devise a demonstrative environment and characterize the causal relationships, thereby facilitating a clear comparison between the two options.

**Questions:**

See above

---

> ### Author Response · Authors · 2024-11-20
> **Responses to Reviewer CaX6**
>
> We appreciate your insightful comments. We will address all your concerns next.
>
> >**W1**: The improvement of HA3C over baselines on the five Mujoco tasks appears to be subtle rather than significant.
>
> Thanks for this point. In Appendix F.4 of the updated version of the paper, we compare HA3C with the best baseline, TD7, with 3M training steps.
> | Algorithm    | Walker2d | HalfCheetah    | Ant | Humanoid    | Hopper |
> | :---        |    :----:   |          ---: |    :----:   |          ---: |          ---: |
> | TD7      | 7570       | 17787  | 9225       | 9850   | 4049  |
> | HA3C   | 8463        | 18687     | 9794   | 11381  |4413   |
>
> From the above results and the learning curves in Appendix F.4  of the updated version of the paper, we can see that the cumulative rewards of HA3C are significantly higher than the cumulative rewards of TD7 on Walker2d, Humanoid, and Hopper.
>
> >**W2**: It would be beneficial to devise a demonstrative environment and characterize the causal relationships, thereby facilitating a clear comparison between the two options.
>
> Thanks for reminding us of this. To illustrate the advantage of HA3C for complex MDP tasks, in Appendix F.1 of the updated version of the paper,  HA3C is compared with No Aug. (HA3C without historical augmentation) on BipedalWalker and BipedalWalker-hardcore tasks. In BipedalWalker a robot is trained to move forward with slightly uneven terrain. Compared with BipedalWalker, BipedalWalker-hardcore is a more complex task, where the above robot is trained to
> move forward with ladders, stumps, and pitfalls. Therefore, the causal relationships in the transitions of BipedalWalker-hardcore are more complex than those in the transitions of BipedalWalker.
>
> | Algorithm      | BipedalWalke | BipedalWalker-hardcore  |
> | :---        |    :----:   |          ---: |
> | HA3C      | 332       | 316   |
> | No Aug.   | 325        |171      |
>
> As we can see, although, both HA3C and No Aug. can get the high cumulative rewards in Bipedal-Walker, only HA3C can get the high cumulative rewards in BipedalWalker-hardcore. This is because by historical augmentation our HA3C can simplify the causal relationships in the transitions of BipedalWalker-hardcore.

---

> > ### Author Response · Authors · 2024-11-26
> > **Sincerely expecting further discussions with Reviewer CaX6**
> >
> > Dear Reviewer CaX6
> >
> > We thank you for helping us improve our paper so far! Your comments are very useful for us to revise the paper to a better version.
> > Based on your comments, we have added the experiments to illustrate the advantage of our historical augmentation in the updated version of the paper. We would be grateful if you could let me know whether our responses address your concerns.
> >
> > Sincerely,
> >
> > Authors

---

### Official Review · Reviewer_nigi · 2024-10-28

**Soundness:** 3
**Presentation:** 3
**Contribution:** 3
**Rating:** 6
**Confidence:** 3

**Summary:**

This paper presents a new algorithm to improve sample efficiency in reinforcement learning by integrating historically augmented states, and presents a series of experiments conducted to validate the effectiveness of this algorithm.

**Strengths:**

1. The examples given in the Section 3 and Appendix B help readers understand the motivation of using historical information.
2. The paper is generally well-organized, making it easy for readers to follow.
3. The proposed method is shown to have strong empirical performance on Mujoco and DMC.

**Weaknesses:**

1. To improve reproducibility, it would be beneficial to supplement the implementation details about the inputs and the parameters of networks such as CNNs used in the encoder.
2. I believe the paper would read more easily after reorganizing the Appendix A and Appendix D, as abbreviations like SkD and MkD may be confusing for those unfamiliar with them.
3. An in-depth analysis of the parameters k and N in the ablation study would greatly enhance readers' understanding of the algorithm. Additionally, I believe more analysis of the running time or the complexity would be helpful, for example, the impact of the parameters k and N on the running time.

Minor comments
1. Is there a mismatch between the Figure 2 and the corresponding description “the dimensionality reduction is only performed on $s_{k−1,t−1}$”?
2. Is there a typo in the results for TD7 on HalfCheetah shown in Table 1 (the reward of 156325)? Additionally, the reward of 45074 for TD3+OFE on Walker2d should also be checked.
3. Formatting:
         a) When the authors or the publication are not part of the sentence, the citation should be in parenthesis by using ‘\citep{}’.
         b) The format for referencing figures should be consistent in the Section 3.

**Questions:**

1. While the proposed historically augmented states can theoretically improve actor-critic methods, could you provide more evidence that demonstrates their applicability to other existing RL methods, aside from the current used TD3?
2. Is the input to HA3C images while the authors mention high-dimensional historical trajectories frequently?

---

> ### Author Response · Authors · 2024-11-20
> **Responses to Reviewer nigi**
>
> We apricate your insightful comments and your positive support and for giving us good scores. We will address all your concerns next.
>
> >**W1**: To improve reproducibility, it would be beneficial to supplement the implementation details about the inputs and the parameters of networks such as CNNs used in the encoder.
>
> Thanks for reminding us of this. We have added the above implementation details in Appendix E of the updated version of the paper.
>
>
> >**W2**: I believe the paper would read more easily after reorganizing Appendix A and Appendix D, as abbreviations like SkD and MkD may be confusing for those unfamiliar with them.
>
> Thanks for reminding us of this.  There is an added table to summarize the abbreviations of different types of policies in Appendix A of the updated version of the paper. The other abbreviations are introduced in Section 2.
>
> >**W3**: An in-depth analysis of the parameters k and N in the ablation study would greatly enhance readers' understanding of the algorithm. Additionally, I believe more analysis of the running time or the complexity would be helpful, for example, the impact of the parameters k and N on the running time.
>
> Thanks for reminding us of this. The analysis of $k$ and $N$ is shown in Section 5.3. The running time of different
> algorithms is shown in  Section 5.4. The running time with different settings of $k$ and $N$ is as follows.
> | $k$    | 6 | 12   | 18   | 24 |
> | :---        |    :----:   |          ---: |    :----:   |          ---: |
> |       | 4h17min      | 4h25min  | 4h36min        | 4h50min    |
>
> | $N$    | 8 | 16   | 64   | 256 |
> | :---        |    :----:   |          ---: |    :----:   |          ---: |
> |       | 4h17min      | 4h22min  | 4h27min        | 4h30min    |
>
> As we can see, the running time of HA3C is not sensitive to $k$ and $N$.
>
> >**M1**: Is there a mismatch between Figure 2 and the corresponding description “the dimensionality reduction is only performed on $s_{k-1,t_1}$”?
>
> Thanks for your question. When predicting $s_{t+1}$ by $s_{k-1,t}$, the dimensionality reduction is only performed on $s_{k-1,t-1}$. To keep the Markov assumption of MDPs, our representation learning does not compress $s_t$.
>
> >**M2**: Is there a typo in the results for TD7 on HalfCheetah shown in Table 1 (the reward of 156325)? Additionally, the reward of 45074 for TD3+OFE on Walker2d should also be checked. **M3:** Formatting: a) When the authors or the publication are not part of the sentence, the citation should be in parenthesis by using ‘\citep{}’. b) The format for referencing figures should be consistent in Section 3.
>
> Thanks for reminding us of this. In the updated version of the paper, we have fixed the errors in the Tables. We also have fixed the format errors.
>
> >**Q1**: While the proposed historically augmented states can theoretically improve actor-critic methods, could you provide more evidence that demonstrates their applicability to other existing RL methods, aside from the current used TD3?
>
> Thanks for this point. In Appendix F.5 of  the updated version of the paper, we combine our historical representation learning with SAC to construct HA3CSAC method.
> Then we evaluate HA3C-SAC on three MuJoCo control tasks including Walker2d, Humanoid, and Hopper. The compared methods includes the original SAC and SALE-SAC, which combines the representation learning with SAC without historical augmentation.
>
> | Algorithm    | Walker2d | Humanoid    | Hopper |
> | :---        |    :----:   |          ---: |    :----:   |
> | SAC   | 3921       | 5498     | 3422  |
> | SALE-SAC      | 6021     | 8368  | 4038       |
> | HA3C   |6950        | 9047     | 4131   |
>
> As we can see, HA3C-SAC outperforms SAC and SALE-SAC on the three Mujoco control tasks.
>
> >**Q2**: Is the input to HA3C images while the authors mention high-dimensional historical trajectories frequently?
>
> The input to HA3C is not images. Each state in HA3C is a vector.
> However, in HA3C, we need to consider all states in a historical trajectory together.
> Therefore, we regard the historical trajectories as high-dimensional data.

---

> ### Comment · Reviewer_nigi · 2024-11-25
>
> I thank authors for the responses. However, I still have two questions about M1:
>
> Q1: In Figure 2, should $s_{t-k, t-2}$ and $s_{t-k+1, t-1}$ instead be $s_{k-1, t-2}$ and $s_{k-1, t-1}$, as you mentioned that “the dimensionality reduction is only performed on $s_{k-1, t-1}$”? This part is a bit confusing to me.
>
> Q2: In Sec. 4.1, you stated, “If $t < k$, one can set each $s_i \in s_{t-k, -1}$ as the zero vector.” Should it instead be $s_{k-t, -1}$ to represent the set of $s_t$ for $t < 0$?

---

> ### Author Response · Authors · 2024-11-25
> **Thank you!**
>
> Dear Reviewer nigi,
>
> We appreciate your feedback. Your concerns are addressed next.
> >**Q1:** In Figure 2, should $s_{t-k,t-2}$  and $s_{t-k+1,t-1}$ instead be $s_{k-1,t-2}$ and $s_{k-1,t-1}$, as you mentioned that “the dimensionality reduction is only performed on $s_{k-1,t-1}$”? This part is a bit confusing to me.
>
> Thanks for reminding us of this.
> We are sorry for this mistake.
>  We have changed $s_{t-k,t-2}$  and $s_{t-k+1,t-1}$ to $s_{k-1,t-2}$ and $s_{k-1,t-1}$ in the Fig.2 of the updated version of the paper.
>
> >**Q2:** In Sec. 4.1, you stated, “If $t<k$, one can set each  $s_t \in s_{t-k,-1}$ as the zero vector.” Should it instead be $s_{k-t,-1}$ to represent the set of $s_t$ for $t<0$?
>
> Thanks for reminding us of this.
> We are sorry for this mistake.
> We have changed $s_{t-k,-1}$ to $s_{k-t,-1}$ in the updated version of the paper.
>
> **Again, thank you so much for the detailed and very constructive comments! Please let us know if you have any further question.**

---

### Official Review · Reviewer_M2GK · 2024-10-31

**Soundness:** 3
**Presentation:** 2
**Contribution:** 3
**Rating:** 5
**Confidence:** 3

**Summary:**

This paper provides an interesting viewpoint on learning a policy that not only depends on the current state but also on the history in Markov decision processes. The key motivation is that, by conditioning on the history, the underlying pattern may be simpler than only conditioning on the current state. The Fibonacci example nicely demonstrates this point. Later, the authors identified two challenges when we want to learn a policy depending on the history: 1) How to ensure we learn a simple pattern based on the history? 2) How to avoid overfitting to the high-dimensional historical data? The core solution proposed by this paper is to learn two encoders - one is used to compress the history into a low-dimensional embedding and the other serves as a latent world model to predict the embedding of the next state based on the action.

**Strengths:**

The overall idea is novel but sound. The proposed method is reasonable from an intuitive perspective.

**Weaknesses:**

- While the Fibonacci example is persuasive, I do not quite understand the example provided in Figure 1. Why the causal function in Figure 1 (b) is simpler than the causal function in Figure 1 (a)? Or this is just a illustrative figure for historical augmentation but not for providing a solid sample? If this is the case, I would like to see a less artificial example to demonstrate on this point (depending on the history can lead to a simpler causal relationship).
- After I go through the algorithmic design of HA3C, I feel this algorithm is closer to “representation learning for RL” such as Dreamer. (By the way, Dreamer lacks citation in L128.) HA3C essentially learns encoders that can compress the state (plus history) in to a latent space and learns an additional latent world model (g) that can predict the dynamics in this latent space. From this perspective, HA3C would better to also compare with other “representation learning for RL” baselines.
- On the experimental results, the improvement of HA3C is marginal over the baseline algorithms on MuJoCo control tasks. From my point of view, this does not indicate that HA3C is ineffective. I think the benefit of HA3C relies on the structure of the problem: the causal relationship based on only the current state is complex but the causal relationship based on the history can be simple. MuJoCo tasks may not be good environment for HA3C. I strongly suggest the authors to find other tasks (or even artificial tasks to demonstrate the effectiveness of HA3C).

**Questions:**

- I do not quite understand the significance of the point demonstrated in Figure 5. Do I understand correctly? HA3C has more points in the red circle. This indicates that HA3C can reach the high-rewarding states more often (or robustly). This information seems a little  duplicated to the training curves demonstrated in Figure 6 or Table 1.
- In L171, should the formula depend on $s_t$ but not $t$ since we are talking about predicting $s_{t+1}$ from $s_t$ but not $t$.
- The citation format is incorrect.

---

> ### Author Response · Authors · 2024-11-20
> **Responses to Reviewer M2GK (Part 1)**
>
> We appreciate your insightful comments. We will address all your concerns next.
>
> >**W1：** While the Fibonacci example is persuasive, I do not quite understand the example provided in Figure 1. Why the causal function in Figure 1 (b) is simpler than the causal function in Figure 1 (a)? Or this is just a illustrative figure for historical augmentation but not for providing a solid sample? If this is the case, I would like to see a less artificial example to demonstrate on this point (depending on the history can lead to a simpler causal relationship).
>
> Thanks for reminding us of this. Figure 1 is an illustrative figure. A theoretical analysis on this figure is shown in Appendix D.4 of the updated version of the paper. **There is another example to demonstrate the advantage of historical augmentation.**
>
> For an object moving in a physics simulation, the state includes time $t$,  its velocity $v$, its mass $m$, and all other physical characteristics. If we need to calculate the acceleration of this object, without considering historical information, we have to perform a complex force analysis. Fortunately, when considering history, the acceleration can be computed by $\Delta v/ \Delta t$.
>
> >**W2：** After I go through the algorithmic design of HA3C, I feel this algorithm is closer to “representation learning for RL” such as Dreamer. (By the way, Dreamer lacks citation in L128.) HA3C essentially learns encoders that can compress the state (plus history) in to a latent space and learns an additional latent world model (g) that can predict the dynamics in this latent space. From this perspective, HA3C would better to also compare with other “representation learning for RL” baselines.
>
> Thanks for reminding us of this. We have cited Dream [1] in L128 of the updated version of the paper. In the first version of the paper, we have already compared our HA3C with **TD7 [2] and TD3-OFE[3], which perform the representation learning in RL**.  In the updated version of the paper, SALE-SAC[2], another RL method with representation learning,  is added to our experiment (see Appendix F.5). Dreamer is proposed to deal with image features in POMDPs, however, HA3C is proposed to improve the sample efficiency in MDPs.
> Therefore Dreamer is not compared with HA3C in our paper.
>
> >**W3：** On the experimental results, the improvement of HA3C is marginal over the baseline algorithms on MuJoCo control tasks. From my point of view, this does not indicate that HA3C is ineffective. I think the benefit of HA3C relies on the structure of the problem: the causal relationship based on only the current state is complex but the causal relationship based on the history can be simple. MuJoCo tasks may not be good environment for HA3C. I strongly suggest the authors to find other tasks (or even artificial tasks to demonstrate the effectiveness of HA3C).
>
> Thanks for this point.
> In Appendix F.4 of the updated version of the paper, we compare HA3C with the best baseline, TD7, with 3M training steps.
> | Algorithm    | Walker2d | HalfCheetah    | Ant | Humanoid    | Hopper |
> | :---        |    :----:   |          ---: |    :----:   |          ---: |          ---: |
> | TD7      | 7570       | 17787  | 9225       | 9850   | 4049  |
> | HA3C   | 8463        | 18687     | 9794   | 11381  |4413   |
>
> From the above results and the learning curves in Appendix F.4, we can see that the cumulative rewards of HA3C are significantly higher than the cumulative rewards of TD7 on Walker2d, Humanoid, and Hopper.
>
> **What's more,**  in Appendix F.1 of the updated version of the paper,  HA3C is compared with No Aug. (HA3C without historical augmentation) on BipedalWalker and BipedalWalker-hardcore tasks. In BipedalWalker a robot is trained to move forward with slightly uneven terrain. Compared with BipedalWalker, BipedalWalker-hardcore is a more complex task, where the above robot is trained to
> move forward with ladders, stumps, and pitfalls. Therefore, the causal relationships in the transitions of BipedalWalker-hardcore are more complex than those in the transitions of BipedalWalker.

---

> ### Author Response · Authors · 2024-11-20
> **Responses to Reviewer M2GK (Part 2)**
>
> | Algorithm      | BipedalWalke | BipedalWalker-hardcore  |
> | :---        |    :----:   |          ---: |
> | HA3C      | 332       | 316   |
> | No Aug.   | 325        |171      |
>
> As we can see, although, both HA3C and No Aug. can get the high cumulative rewards in Bipedal-Walker, only HA3C can get the high cumulative rewards in BipedalWalker-hardcore. This is because by historical augmentation our HA3C can simplify the causal relationships in the transitions of BipedalWalker-hardcore.
>
> >**Q1:** I do not quite understand the significance of the point demonstrated in Figure 5. Do I understand correctly? HA3C has more points in the red circle. This indicates that HA3C can reach the high-rewarding states more often (or robustly). This information seems a little duplicated to the training curves demonstrated in Figure 6 or Table 1.
>
> Yes, your understanding is correct. Figure 5 explains the advantage of historical augmentation from the perspective of data manifold. As we can see, the data distributions in the red circles are significant differences in different subfigures.
>
>
> >**Q2:** In L171, should the formula depend on $s_t$ but not $t$ since we are talking about predicting $s_{t+1}$ from $s_{t}$.
>
> Thanks for this point. We can define
> $$
> f(t) =\frac{1} {\sqrt{5}} \left[\left(\frac{1+\sqrt{5}}{2}\right)^{t} -\left(\frac{1-\sqrt{5}}{2}\right)^{t} \right].
> $$
> Then the formula  in L171 can be written as
> $$
>  s_{t+1}  =\frac{1} {\sqrt{5}} \left[\left(\frac{1+\sqrt{5}}{2}\right)^{f^{-1}(s_t)+1} -\left(\frac{1-\sqrt{5}}{2}\right)^{f^{-1}(s_t)+1} \right].
> $$
>
> >**Q3:** The citation format is incorrect.
>
> Thanks for reminding us of this. We have fixed this error in the updated version of the paper.
>
>
> [1] Danijar Hafner, Timothy Lillicrap, Jimmy Ba, and Mohammad Norouzi. Dream to control: Learning
> behaviors by latent imagination. In International Conference on Learning Representations, 2019a.
>
> [2] Scott Fujimoto,Wei-Di Chang, Edward J Smith, Shixiang Shane Gu, Doina Precup, and David Meger.
> For SALE: State-action representation learning for deep reinforcement learning. In Thirty-seventh
> Conference on Neural Information Processing Systems, 2023.
>
> [3]  Kei Ota, Tomoaki Oiki, Devesh Jha, Toshisada Mariyama, and Daniel Nikovski. Can increasing input
> dimensionality improve deep reinforcement learning? In International Conference on Machine
> Learning, pp. 7424–7433. PMLR, 2020.

---

> ### Author Response · Authors · 2024-11-26
> **Sincerely expecting further discussions with Reviewer M2GK**
>
> Dear Reviewer M2GK
>
> We thank you for helping us improve our paper so far! Your comments are very useful for us to revise the paper to a better version.
> We hope our responses have adequately addressed your concerns  and would greatly appreciate any further feedback you may have.
>
> Sincerely,
>
> Authors

---

### Official Review · Reviewer_uKeS · 2024-11-03

**Soundness:** 2
**Presentation:** 3
**Contribution:** 2
**Rating:** 3
**Confidence:** 4

**Summary:**

For deep reinforcement learning, the paper proposes to augment the state with compressed historical to improve sample efficiency and performance. Some theoretical analysis provides optimality and convergence properties for the proposed state augmentation when certain conditions are satisfied. Numerical experiments show decent performance for the proposed method compared to some existing methods.

**Strengths:**

- Having an appropriate representation is important for RL agents. For problems with the Markov property, it is typically for a RL agent to consider only the current state as the state is known to be sufficient to make optimal decisions. The proposed idea to augment history to help the agent to improve its representation learning is an very interesting ideas and sounds promising from simple examples as discussed.

- The proposed method shows decent performance in numerical experiments, and the effectiveness of history augmentation is numerically illustrated in the ablation study.

**Weaknesses:**

- Although the paper provides some analysis for optimality and convergence properties of the proposed provided, these properties do not provide any insight into why the history augmentation helps representation learning. And there is no analysis on potential sample complexity reduction. Since it is assumed that the state is kept and completely uncompressed in the encoder output, most of the results are expected. It is likely that simpler arguments may be available by arguing that the original state-dependent optimal policy is also a feasible policy with the augmentation.

- In both the analysis of Section 4.1 and the algorithm design in Section 4.2, it is not clear whether we non trivial augmentation is needed. For example, the analysis seems to completely go through when $f(s_{k, t}) = s_t$ in Section 4.1, and nothing seems to prevent the HA3C algorithm to ignore the history and ending up getting $z^{s^{k, t}_\alpha} = s_t$.

- Although the ablation study shows better performance with the proposed history augmentation, the improvement does not seem significant given those largely overlapping confidence areas. Additional experiments like showing how performance varies by varying the length of the history augmentation may provide some trends that could be more convincing.

- Comparing to existing methods, the proposed seem to perform decently in the numerical experiments, but the improvement is not that significant especially compared with TD7. Without additional analysis on the quality of the learned representation, it is not clear if the performance benefits indeed come from the proposed history augmentation.

**Questions:**

- Beyond the simple examples, are there theoretical or numerical analysis showing sample complexity benefits with history augmentation?

---

> ### Author Response · Authors · 2024-11-19
> **Responses to Reviewer uKeS (Part 1)**
>
> We thank you for giving us detailed and very helpful comments (strengths and concerns). Your concerns are addressed next.
> >**W1.1：** Although the paper provides some analysis for optimality and convergence properties of the proposed provided, these properties do not provide any insight into why the history augmentation helps representation learning. And there is no analysis on potential sample complexity reduction. Since it is assumed that the state is kept and completely uncompressed in the encoder output, most of the results are expected.
>
> Thanks for reminding us of this. In the updated version of the paper, we show the theoretical analysis of sample complexity reduction from historical augmentation in Appendix D.4. This theoretical analysis is based on the following two facts:
> 1) Historical augmentation can improve exploration in DRL. The policy can generate different actions for different transition trajectories that end with the same state.
>  2) Historical augmentation also can improve exploitation in DRL.
> A high-reward action from the added random noise may be hard to regenerate, because the added noise is independent of the parameters of the policy network. History augmentation may simplify the causal relationships between the states and the explored high-reward actions, thus the policy network can effectively learn and then regenerate these actions.
>
> Our representation learning does not compress the current state to ensure obtaining the optimal policy. The state abstraction (compression) for the historical states in our representation learning can improve the sample efficiency in the optimization
> （See the analysis in Section 5.3).
> >**W1.2：** It is likely that simpler arguments may be available by arguing that the original state-dependent optimal policy is also a feasible policy with the augmentation.
>
> There are four subsections in our theoretical analysis (Appendix D).
> In Appendix D.1, we prove that the original state-dependent optimal policy is also a feasible policy with the augmentation.
> In Appendix D.2, we analyze the convergence of our method.
> In Appendix D.3, we give the approximation error.
> In Appendix D.4, we analyze the sample complexity reduction from our historical augmentation.
>
> >**W2：**  In both the analysis of Section 4.1 and the algorithm design in Section 4.2, it is not clear whether we non-trivial augmentation is needed.  For example, the analysis seems to completely go through when $f(s_{k,t})=s_t$ in Section 4.1, and nothing seems to prevent the HA3C algorithm to ignore the history and ending up getting $z^{s_{k,t}}_{\alpha}$.
>
> When $k=1$, i.e., $(s_{k,t})=s_t$ , our algorithm still converges. The necessity of historical augmentation is analyzed in Fig. 5 of Section 4.2. This figure illustrates that history augmentation may simplify the causal relationships between the states and the explored high-reward actions, thus the policy network can effectively learn and then regenerate these actions.
> We also added the theoretical analysis of sample complexity reduction from historical augmentation in Appendix D.4 of the updated version of the paper.
>
> >**W3：** Although the ablation study shows better performance with the proposed history augmentation, the improvement does not seem significant given those largely overlapping confidence areas. Additional experiments like showing how performance varies by varying the length of the history augmentation may provide some trends that could be more convincing.
>
> Thanks for reminding us of this. The experiments in Section 5.3 has already shown how performance varies by varying the length of the history augmentation. To illustrate the advantage of HA3C for complex MDP tasks, in Appendix F.1 of the updated version of the paper, we test HA3C and No Aug. (HA3C without historical augmentation) on BipedalWalker and BipedalWalker-hardcore tasks.
> | Algorithm      | BipedalWalke | BipedalWalker-hardcore  |
> | :---        |    :----:   |          ---: |
> | HA3C      | 332       | 316   |
> | No Aug.   | 325        |171      |
>
> As we can see, although, both HA3C and No Aug. can get the high cumulative rewards in Bipedal-Walker, only HA3C can get the high cumulative rewards in BipedalWalker-hardcore. This is because by historical augmentation our HA3C can simplify the causal relationships in the transitions of BipedalWalker-hardcore.

---

> ### Author Response · Authors · 2024-11-19
> **Responses to Reviewer uKeS (Part 2)**
>
> >**W4：** Compared to existing methods, the proposed seem to perform decently in the numerical experiments, but the improvement is not that significant especially compared with TD7. Without additional analysis on the quality of the learned representation, it is not clear if the performance benefits indeed come from the proposed history augmentation. **Q1：** Beyond the simple examples, are there theoretical or numerical analysis showing sample complexity benefits with history augmentation?
>
>
>  We have added the following analysis to illustrate that sample complexity indeed benefits with history augmentation.
> 1) The theoretical analysis in  Appendix D.4 (See the responses of W1)
> 2)  The experiment on BipedalWalker and BipedalWalker-hardcore tasks (See the responses of W3)
> 3)  Longer training runs
>
> In Appendix F.4, we compare HA3C with the best baseline, TD7, with 3M training steps.
> | Algorithm    | Walker2d | HalfCheetah    | Ant | Humanoid    | Hopper |
> | :---        |    :----:   |          ---: |    :----:   |          ---: |          ---: |
> | TD7      | 7570       | 17787  | 9225       | 9850   | 4049  |
> | HA3C   | 8463        | 18687     | 9794   | 11381  |4413   |
>
> From the above results and the learning curves in Appendix F.4, we can see that the cumulative rewards of HA3C are significantly higher than the cumulative rewards of TD7 on Walker2d, Humanoid, and Hopper.
>
> 4) Combining Historical Representation Learning with SAC
>
> In Appendix F.5, we combine our historical representation learning with SAC to construct HA3CSAC.
> Then we evaluate HA3C-SAC on three MuJoCo control tasks including Walker2d, Humanoid, and Hopper. The compared methods includes the original SAC and SALE-SAC, which combines the representation learning with SAC without historical augmentation.
>
> | Algorithm    | Walker2d | Humanoid    | Hopper |
> | :---        |    :----:   |          ---: |    :----:   |
> | SAC   | 3921       | 5498     | 3422  |
> | SALE-SAC      | 6021     | 8368  | 4038       |
> | HA3C   |6950        | 9047     | 4131   |
>
> As we can see, HA3C-SAC outperforms SAC and SALE-SAC on the three Mujoco control tasks.
>
>
> **Furthermore**, the analysis in Fig.5 of Section 4.2 also can illustrate the advantage of our historical representation learning.

---

> ### Author Response · Authors · 2024-11-26
> **Sincerely expecting further discussions with Reviewer uKeS**
>
> Dear Reviewer uKeS
>
> We thank you for helping us improve our paper so far!
> Your comments are very useful for us to revise the paper to a better version.
>
> Based on your comments, we have added a lot of experiments and theoretical analysis in the updated version of the paper.
> We would be grateful if you could let me know whether our responses address your concerns.
>
> Sincerely,
>
> Authors

---

### Author Response · Authors · 2024-11-20
**Thank you and hope you may take a quick look at our responses.**

Dear Reviewers,

Thank you so much for spending your precious time reviewing our paper. We have received very informative feedback from your comments. They are very useful for us to revise the paper to a better version.

Meanwhile, as you may notice, we have responded to all of you. We are now writing simply to wonder if you could spend a few minutes taking a quick look at our responses. Also, based on our responses, if you think there should be more experiments/analyses, please let us know asap so that we can have enough time to finish them early.

We would also really appreciate it if you could give us a little more feedback based on our responses (e.g., do our responses resolve your concerns?)

Sincerely,

Authors

---

### Author Response · Authors · 2024-11-24
**Sincerely expecting further discussions with reviewers**

Dear Reviewers,

We thank you for helping us improve our paper so far! The ICLR author-reviewer discussion deadline (26 Nov 2024) is approaching. We genuinely hope to have a further discussion with you. We have carefully read your comments and tried our best to address your concerns. We would be grateful if you could let me know whether our responses address your concerns.

Sincerely,

Authors

---

### Meta-Review · Area_Chair_1mZ6 · 2024-12-23

**Metareview:**

### Summarization
This paper aims to use historical state information to improve the current RL algorithms. The motivation arises from the fact that historical information may makes problem easier to solve. Then, the authors use a CNN to map high-dimensional data into low-dimensional representation to avoid overfitting caused by high-dimensional data. Finally, they apply an adapted TD3 algorithm to the low-dimensional representation, propose an algorithm named HA3C, and conduct some experiments to illustrate the performance of the algorithm.

###  Strengths
* The paper is well structured and clearly written
* The proposed algorithm is quite intuitive

### Weaknesses
* The improvement of the algorithm in the experiments is not significant
* The explaination of why history augmentation helps and how it reduce sample complexity is not clear
* The motivation for using historical information is not entirely convincing

Overall, the weaknesses of this paper outweigh the strengths, so I tend to reject.

**Additional Comments On Reviewer Discussion:**

* Reviewer CaX6, Reviewer uKeS, Reviewer M2GK challenged the significance of the improvement in the experiments, and the authors provide extensive experiments in Appendix F.

* Reviewer M2GK questioned why the history augmentation helps and how it reduce sample complexity and the authors added this part in Appendix D.4.

---

### Decision · Program_Chairs · 2025-01-22

Reject